# The TBC1D31/praja2 complex controls primary ciliogenesis through PKA-directed OFD1 ubiquitylation

Emanuela Senatore[1,†], Francesco Chiuso[1,†], Laura Rinaldi[1], Daniela Intartaglia[2], Rossella Delle Donne[1], Emilia Pedone[3], Bruno Catalanotti[4], Luciano Pirone[3], Bianca Fiorillo[4], Federica Moraca[4,5], Giuliana Giamundo[2], Giovanni Scala[6], Andrea Raffeiner[7,8], Omar Torres-Quesada[7,8,9], Eduard Stefan[7,8,9], Marcel Kwiatkowski[7], Alienke van Pijkeren[7], Manuela Morleo[2] (iD), Brunella Franco[2,10] (iD), Corrado Garbi[1], Ivan Conte[2,6] (iD) & Antonio Feliciello[1,*] (iD)

## Abstract

The primary cilium is a microtubule-based sensory organelle that dynamically links signalling pathways to cell differentiation, growth, and development. Genetic defects of primary cilia are responsible for genetic disorders known as ciliopathies. Orofacial digital type I syndrome (OFDI) is an X-linked congenital ciliopathy caused by mutations in the OFD1 gene and characterized by malformations of the face, oral cavity, digits and, in the majority of cases, polycystic kidney disease. OFD1 plays a key role in cilium biogenesis. However, the impact of signalling pathways and the role of the ubiquitin-proteasome system (UPS) in the control of OFD1 stability remain unknown. Here, we identify a novel complex assembled at centrosomes by TBC1D31, including the E3 ubiquitin ligase praja2, protein kinase A (PKA), and OFD1. We show that TBC1D31 is essential for ciliogenesis. Mechanistically, upon G-protein-coupled receptor (GPCR)-cAMP stimulation, PKA phosphorylates OFD1 at ser735, thus promoting OFD1 proteolysis through the praja2-UPS circuitry. This pathway is essential for ciliogenesis. In addition, a non-phosphorylatable OFD1 mutant dramatically affects cilium morphology and dynamics. Consistent with a role of the TBC1D31/praja2/OFD1 axis in ciliogenesis, alteration of this molecular network impairs ciliogenesis in vivo in Medaka fish, resulting in developmental defects. Our findings reveal a multifunctional transduction unit at the centrosome that links GPCR signalling to ubiquitylation and proteolysis of the ciliopathy protein OFD1, with important implications on cilium biology and development. Derangement of this control mechanism may underpin human genetic disorders.

**Keywords** OFD1; PKA; praja2; primary cilium; ubiquitin
**Subject Categories** Membranes & Trafficking; Post-translational Modifications & Proteolysis; Signal Transduction
**The EMBO Journal (2021) 40: e106503**

## Introduction

The primary cilium is the principal sensory organelle in most eukaryotic cells and represents an intracellular hub where distinct signalling networks integrate and focus. A variety of receptors, scaffolds, ion channels, adaptor molecules and effector enzymes are localized within the ciliary compartment, playing a central role in development, metabolism, growth and differentiation (Ishikawa & Marshall, 2011; Oh *et al*, 2015; Hilgendorf *et al*, 2016; Nachury & Mick, 2019). Under growth-arrested conditions, the mother centriole of the centrosome migrates to the cell surface and starts to elongate as axonemal structure composed of nine doublet microtubules enveloped within the plasma membrane, forming the mature cilium and its basal body. This is an evolutionary conserved and highly

1 Department of Molecular Medicine and Medical Biotechnologies, University Federico II, Naples, Italy
2 Telethon Institute of Genetics and Medicine, Pozzuoli, Italy
3 Institute of Biostructures and Bioimaging, CNR, Naples, Italy
4 Department of Pharmacy, University Federico II, Naples, Italy
5 Net4Science srl, University "Magna Græcia", Catanzaro, Italy
6 Department of Biology, University Federico II, Naples, Italy
7 Institute of Biochemistry, University of Innsbruck, Innsbruck, Austria
8 Center for Molecular Biosciences, University of Innsbruck, Innsbruck, Austria
9 Tyrolean Cancer Research Institute, Innsbruck, Austria
10 Department of Translational Medical Science, University Federico II, Naples, Italy
*Corresponding author. Tel: +39 081 7463615; E-mail: feliciel@unina.it
†These authors contributed equally to this work

regulated process controlled by a variety of pericentriolar proteins, regulators and scaffolds, all of which contribute to the formation and maintenance of the ciliary morphology and structure (Sanchez & Dynlacht, 2016). Mutations affecting genes involved in the dynamic control of cilium biogenesis often cause developmental genetic disorders, also known as ciliopathies (Valente *et al*, 2014; Reiter & Leroux, 2017). The OFD1 gene encodes a component of the centrosome/basal body and pericentriolar satellites that plays a major role in cilium biogenesis (Lopes *et al*, 2011). Germline inactivating mutations of OFD1 cause the Oral-Facial-Digital type I (OFDI) syndrome, a developmental disorder usually characterized by typical oral-facial-digital malformations, renal cystic disease and central nervous system involvement (Macca & Franco, 2009; Bruel *et al*, 2017). In serum-deprived cells, removal of OFD1 from centriolar satellites through the autophagy machinery is required for the onset of ciliogenesis (Tang *et al*, 2013). A role for OFD1 in non-ciliary pathways has also been reported (Abramowicz *et al*, 2017; Iaconis *et al*, 2017; Alfieri *et al*, 2020). The ubiquitin-proteasome system (UPS) controls the levels of ciliary regulatory proteins, contributing to the dynamic assembly/disassembly of the primary cilium (Kasahara *et al*, 2014; Liu *et al*, 2014; Wheway *et al*, 2015; Kwon & Ciechanover, 2017; Nagai *et al*, 2018; Tsai *et al*, 2019; Wiegering *et al*, 2019). However, the role of signalling enzymes in OFD1-dependent functions at the ciliary compartment and the impact of the ubiquitin-proteasome system (UPS) on OFD1 stability/activity are largely unknown.

Growing evidence indicates that deregulation of signalling pathways, involving Sonic Hedgehog, Wnt, Notch and cAMP cascades, generated at—or converging to—the ciliary compartment contributes to human disorders, such as developmental deficits, neurodegeneration and cancer (Anvarian *et al*, 2019; Jeng *et al*, 2020). cAMP is an ancient second messenger that controls key biological activities, including metabolism, cell growth, development, differentiation and synaptic activities. Protein Kinase A (PKA) is the main effector of cAMP action and is present in the cell as tetrameric holoenzyme composed of two regulatory (R) and two catalytic (PKAc, C) subunits. Activation of the adenylate cyclase by a GPCR ligand induces a cAMP-mediated dissociation of the PKA holoenzyme and consequent release of active PKAc subunits. Phosphorylation of cellular substrates by PKAc regulates important biological functions (Taylor *et al*, 2013; Newton *et al*, 2016; Rinaldi *et al*, 2019). Compartmentalization of PKA at discrete intracellular sites by A-kinase anchor proteins (AKAPs) contributes to the activation, dissemination and attenuation of cAMP signals at distal sites from signal generation (Yang & McKnight, 2015; Jones *et al*, 2016; Reggi & Diviani, 2017; Rinaldi *et al*, 2017; Torres-Quesada *et al*, 2017; Rinaldi *et al*, 2018; Bucko *et al*, 2019). The AKAP praja2 binds and targets PKA holoenzyme to the cell membrane, perinuclear region and cellular organelles. Co-localization of praja2•PKA complexes with PKA substrate/effector molecules ensures efficient integration and propagation of the locally generated cAMP to distinct target sites (Lignitto *et al*, 2011a). praja2 acts as an E3 ubiquitin ligase that controls ubiquitylation and stability of colocalized signalling enzymes, including PKA, adapter proteins and tumour suppressors (Lignitto *et al*, 2011b; Lignitto *et al*, 2013; Sepe *et al*, 2014; Zhang *et al*, 2015; Rinaldi *et al*, 2016; Song *et al*, 2019). PKA regulates different aspects of cilium biology. Thus, proteomic and functional analyses identified components of the cAMP cascade as residents

and regulators of the ciliary compartment (Mukherjee *et al*, 2016; Siljee *et al*, 2018; Sherpa *et al*, 2020). In this context, activation of PKA within the cilium plays an inhibitory role on the Sonic Hedgehog pathway, a master regulator of embryonic development (Chen *et al*, 2011; Vuolo *et al*, 2015). The identification of orphan receptor GPCR (Gpr161) and adenylate cyclases within the cilium suggested that locally generated cAMP microdomains directly controls the activation of ciliary PKA and the signal dissemination to co-targeted ciliary effector proteins (Mukhopadhyay *et al*, 2013). cAMP signalling also contributes to cilium biogenesis and dynamics (Pal & Mukhopadhyay, 2015; Bachmann *et al*, 2016; Tschaikner *et al*, 2020). A link between PKA signalling and the ubiquitin-proteasome system at ciliary compartment has been recently discovered (Porpora *et al*, 2018). Thus, PKA phosphorylation of NIMA-related kinase NEK10 promotes its ubiquitylation by the E3 ligase CHIP/Stub1. Ubiquitylated NEK10 undergoes proteasomal degradation, leading to primary cilium resorption (Porpora *et al*, 2018). However, the role of the PKA-ubiquitin system in cilium biogenesis and its relevant targets are still unknown.

Here, we report the identification of a novel multifaceted signalling complex assembled at the centrosome by TBC1D31 that finely controls the PKA-mediated phosphorylation of OFD1 and its ubiquitin-dependent proteolysis through the proteasome. Interfering with this control mechanism affects cilium biogenesis and Medaka fish development.

## Results

### TBC1D31 anchors praja2 to centrosome and centriolar satellites

A yeast two-hybrid screening using the C-terminus of praja2 as bait and a human brain cDNA library identified a clone encoding for the C-terminus (residues 940–970) of TBC1D31, a protein with unknown functions that localizes to the centrosome and centriolar satellites(Gupta *et al*, 2015).

First, we asked whether praja2 and TBC1D31 interact in cell lysates. Co-immunoprecipitation (CoIp) experiments confirmed the interaction between praja2 and TBC1D31 (Fig 1A). By deletion mutagenesis and CoIp assays, we identified residues 530–630 as the praja2 segment that binds to TBC1D31 (Fig 1B and C). GST pull-down experiments confirmed that residues 940–970 of TBC1D31 interact with praja2 (Fig 1D). praja2 is known to anchor PKA to specific intracellular sites (Lignitto *et al*, 2011a). Accordingly, we tested whether PKA was present in the praja2/TBC1D31 complex. As suspected, the PKAc subunit, along with praja2, was recovered in the TBC1D31 immunoprecipitates using antibodies raised against residues 239–358 of human TBC1D31 (Fig 1E, Appendix Fig S1A and B). In *situ* immunostaining analysis confirmed that TBC1D31 is localized at the centrosome and centriolar satellites (Fig 1F, upper panels, Appendix Fig S1B). Moreover, a fraction of the praja2 signal colocalizes with GFP-TBC1D31, supporting the presence of a praja2/TBC1D31 complex within the same intracellular compartment (Fig 1F, lower panels). A similar partial immunostaining pattern of praja2 at centrosome was also observed (Fig EV1A). TBC1D31 acts as an anchor for praja2. Thus, the genetic silencing of TBC1D31 dramatically reduced the localization of praja2 at the centrosome and centriolar satellites (Figs 1G and EV1B–D). In contrast, praja2

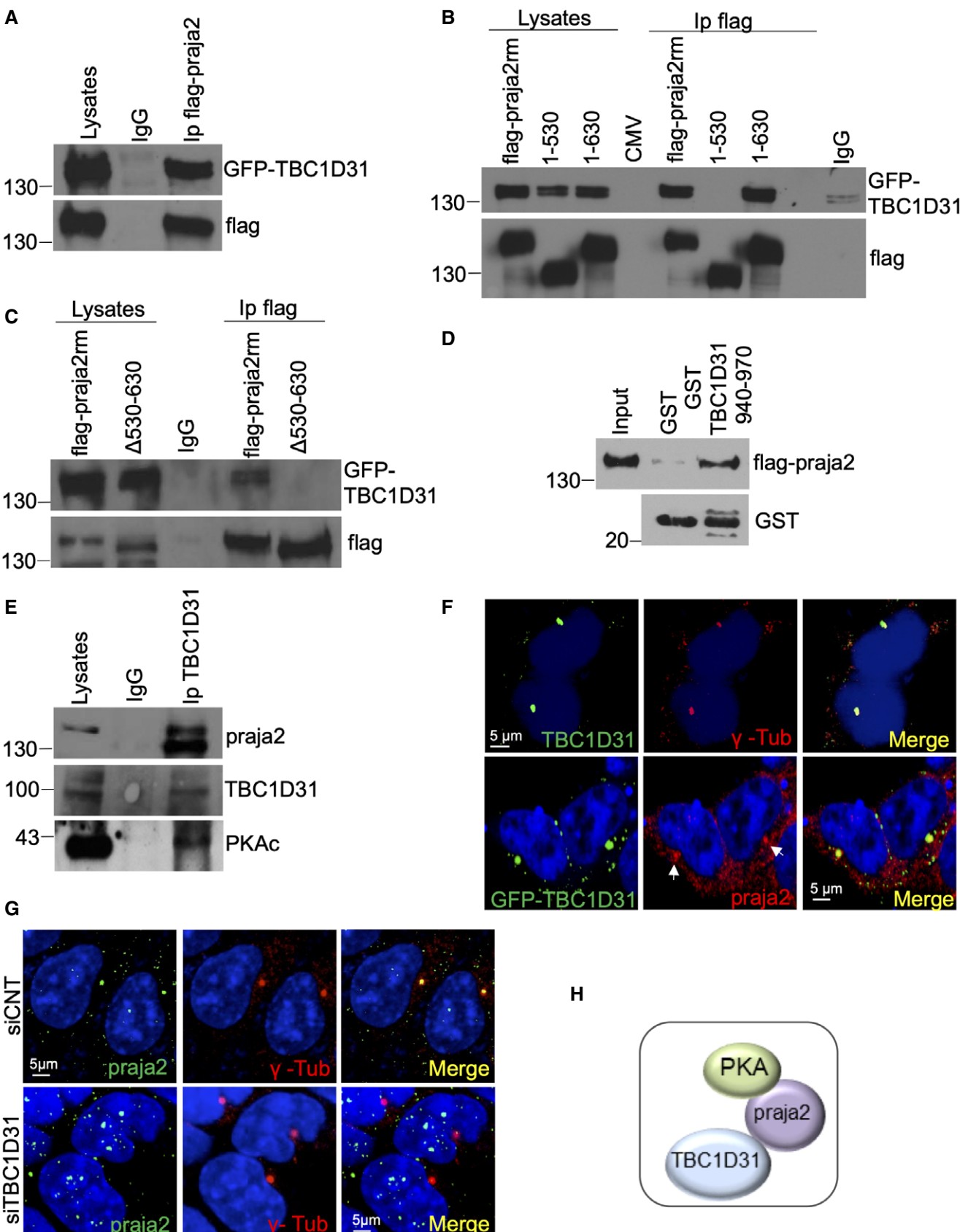

**Figure 1.**

**Figure 1.  TBC1D31 binds and targets praja2 to the centrosome.**

A   Co-immunoprecipitation of flag-praja2 and GFP-TBC1D31 from lysates of HEK293 cells. The immunoprecipitation (Ip) was performed using an anti-flag antibody or control IgG.

B, C   Same as in (A), with the exception that cells expressing flag-praja2rm or praja2 deletion mutants (praja2$_{1-530}$, praja2$_{1-630}$ and praja2$_{\Delta530-630}$) were included in the analysis.

D   Lysates expressing flag-praja2 were subjected to pull down assay with GST and GST-TBC1D31$_{940-970}$ polypeptides.

E   Co-immunoprecipitation of endogenous TBC1D31/praja2/PKAc complex from cell lysates.

F   Staining of HEK293 cells with anti-TBC1D31, anti-γ-tubulin and anti-praja2 antibodies. Nuclei were stained with DRAQ5 (blue). Where indicated, cells were transfected with GFP-TBC1D31. Arrows indicate the pool of praja2 colocalizing with TBC1D31 staining at the centrosome.

G   Cells transfected with control siRNA (siCNT) or siRNA targeting TBC1D31 (siTBC1D31) were stained for praja2, anti-γ-tubulin and DRAQ5.

H   Schematic picture of TBC1D31/praja2/PKA complex.

Source data are available online for this figure.

silencing had no significant impact on the intracellular localization of TBC1D31 (Fig EV1E and F), supporting the model by which TBC1D31 acts as an anchor for praja2 at the centrosome and centriolar satellites (Fig 1H).

**Binding modules of TBC1D31/praja2 complex**

Next, we tested whether praja2 and TBC1D31 interact *in vitro*. A fusion protein carrying residues 531–631 of praja2 fused to the C-terminus of glutathione S-transferase polypeptide (GST) coprecipitated GFP-TBC1D31 from cell lysates (Fig 2A). To identify the minimal core domain on praja2 that binds TBC1D31, we performed *in vitro* microscale thermophoresis binding experiments using partially overlapping synthetic peptides spanning the praja2$_{530-630}$ domain. As shown in Fig 2B, praja2$_{530-570}$ and praja2$_{550-610}$ peptides bind the C-terminus domain of TBC1D31 with micromolar affinity (K$_D$ 37 μM and K$_D$ 80 μM, respectively), whereas no binding was observed with the praja2$_{590-630}$ peptide. This finding suggested that praja2$_{550-570}$ segment contributes to the interaction with TBC1D31. As predicted, deletion of residues 550–570 of praja2 (praja2$_{\Delta550-570}$) dramatically reduced the binding to GFP-TBC1D31 (Fig 2C).

The molecular basis of praja2 and TBC1D31 interaction was investigated by docking and molecular dynamics (MD) studies. praja2$_{550-570}$ (Fig 2B) and TBC1D31$_{941-970}$ (Fig 2D) 3D structures were generated using the threading approach implemented in I-TASSER website. praja2$_{550-570}$ 3D structure resulted mostly coiled, while the TBC1D31$_{941-970}$ domain was modelled as α-helix with a kink at the level of Q941. Accordingly, CD spectra showed that TBC1D31$_{941-970}$ domain assumed a partial helical structure (Fig 2E, Appendix Fig S2A and B). A two-step docking procedure followed by 2 μs classical MD simulations reported a binding mode represented by three main clusters (Appendix Fig S2C and D, Movie EV1). The binding is mainly driven by the arginine-rich stretch R957-R961 (RARHR) of TBC1D31 that establishes cation-π and ionic interactions with the praja2 stretch F553-D558 and with E564. Moreover, only discontinuous interactions of TBC1D31 R948 and R951 residues, mainly with the praja2 D570 residue, were observed (Movie EV1). To validate the proposed binding mode, we designed two different mutants of the C-terminal TBC1D31 peptide: 1. TBC1D31$_{ADA}$ triple mutant (R957A, R959D and H960A) peptide; 2. TBC1D31$_{AA}$ double-mutant (R948A and R951A) peptide. Microscale thermophoresis experiments showed that the interaction between praja2$_{530-570}$ and TBC1D31$_{AA}$ mutant was preserved, whereas it was almost abolished with TBC1D31$_{ADA}$ (Fig 2F, Appendix Fig S3A). In addition, CD spectra showed a partial helical structure for both mutant peptides,

without any appreciable difference with wild-type (Appendix Fig S3B), supporting the MD-derived hypothesis of a specific role of residues R957, R959 and H960 of TBC1D31 in praja2 binding activity.

**OFD1 is a component of the TBC1D31/praja2 complex**

To gain insight into the TBC1D31/praja2 complex and characterize additional relevant partners, we took advantage of the available proteomic databases to build an interaction network for praja2 and TBC1D31 using the GeneMANIA "Physical Interactions" catalogue and the PPI databases as BioGRID and PathwayCommons (Fig 3A). The network was filtered to highlight relationships between first-order interactors, and the protein complexes were functionally annotated for Gene Ontology criteria (Fig 3B). Protein complexes involving protein kinases, including PKA, microtubule-organizing center (MTOC), centrosomes, ciliary proteins, centrioles and pericentriolar satellites were functionally extracted from the network. Orofacial digital syndrome protein 1 (OFD1), an essential regulator of primary ciliogenesis, was identified as a component of the assembled network. This was in agreement with previous observations that OFD1 is part of a PKA interaction network identified in a collection of cancer cells, glioblastoma tissues and lung cancer (Rinaldi *et al*, 2019; Coles *et al*, 2020). First, we sought to demonstrate that OFD1 and TBC1D31 interact in cells. Thus, CoIp assays confirmed that TBC1D31 and OFD1 form a stable complex in cell lysates (Fig 3C). Moreover, a trimeric complex composed of TBC1D31, praja2 and OFD1 could be isolated from cell lysates (Fig 3C) and tissue (Fig 3D and E). *In situ* immunostaining analysis revealed that TBC1D31, OFD1 and praja2 signals partly colocalized, confirming that the three proteins may reside within the same intracellular compartment (Figs 3F and EV2A–C).

**PKA phosphorylates OFD1**

The data above indicate that OFD1 is part of a multimeric complex assembled at the centrosome by TBC1D31, which includes praja2 and PKA. This suggested that OFD1 is a target of PKA. Primary sequence analysis revealed the presence of a PKA consensus site (RRL**S**$_{735}$S) in the human OFD1 protein that is conserved in the mouse ortholog (Fig 4A). An additional PKA site (RRQS$_{899}$) was present only in the human variant. First, we attempted to prove that OFD1 is a direct target of PKA. To this end, we performed *in vitro* phosphorylation assays using purified recombinant his-tagged PKAc and GST-fused polypeptides spanning the C-terminal segment of human OFD1 (OFD1$_{664-1,012}$), either wild-type or the double-mutant (S735A, S899A). The results show that GST-OFD1$_{664-1,012}$ was heavily

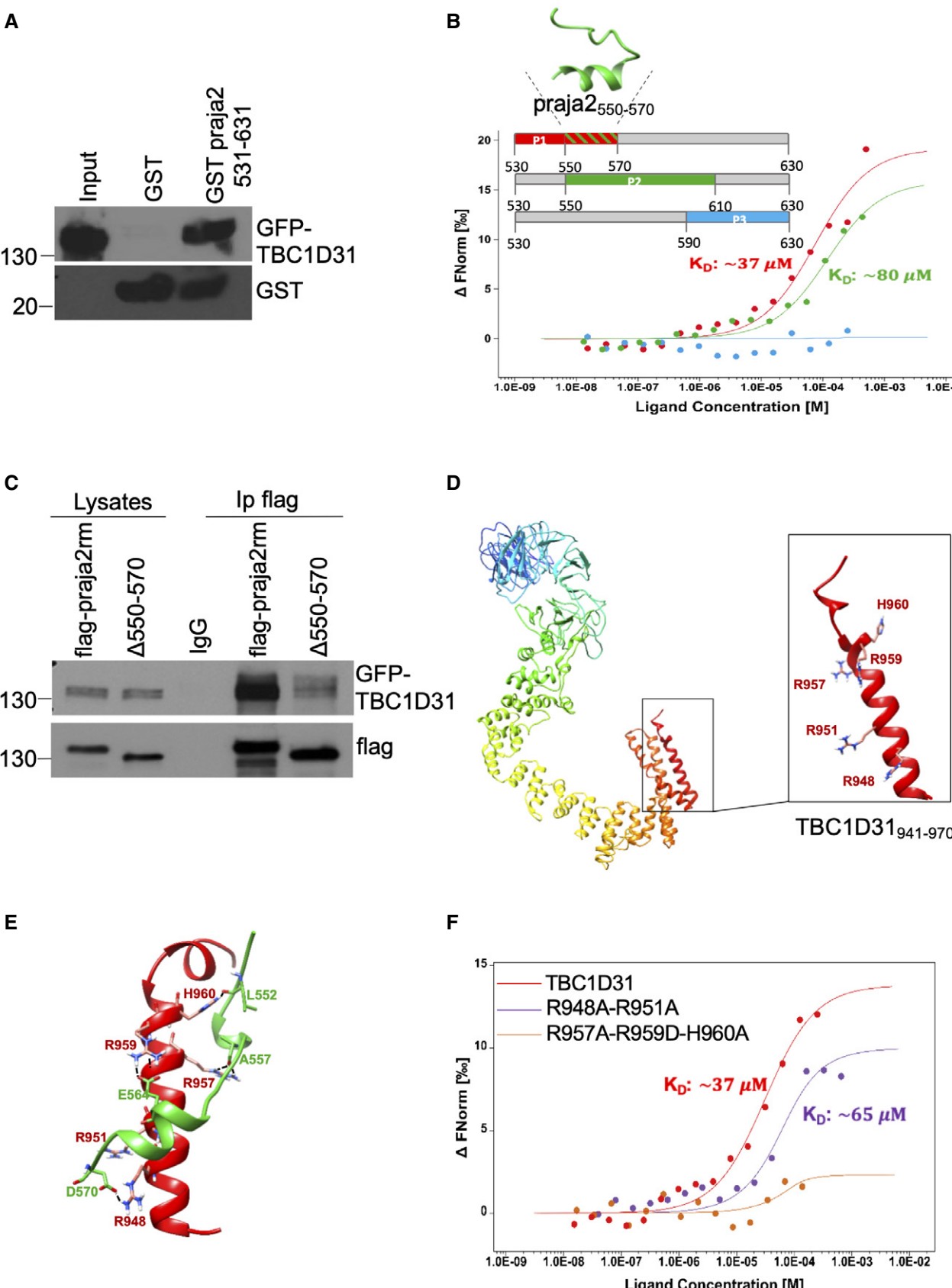

**Figure 2.**

◄

**Figure 2. Modelling TBC1D31/praja2 binding *in vitro*.**

A Lysates from HEK293 cells expressing GFP-TBC1D31 were subjected to pull down assay with GST and GST-praja2$_{531-631}$ polypeptides.

B MST signal (normalized fluorescence) of P1 (red curve), P2 (green curve) and P3 (cyan curve) plotted against TBC1D31, at increasing concentrations of peptides. The threading modelled structure of the overlapping binding segment of praja2 (praja2$_{550-570}$) is shown.

C Co-immunoprecipitation of GFP-TBC1D31 and flag-praja2 ring mutant (flag-praja2rm) or praja2$_{\Delta550-570}$.

D Threading modelled structure of TBC1D31, with a zoom of its C-terminus. Mutated residues are highlighted in stick coloured by atom type.

E MD derived binding mode of praja2$_{530-570}$ (green cartoon) to the C-terminal region of TBC1D31 (red cartoon).

F MST signal of P1 plotted against increasing concentrations of TBC1D31 peptides: wild-type (red curve), R948A-R951A (violet curve) and R957-R959D-H960A (orange curve).

Source data are available online for this figure.

phosphorylated by PKAc (Fig 4B). Mutations of S735 and S899 to alanine (S735A, S899A) completely abrogated phosphorylation of OFD1 segment by PKAc. PKAc autophosphorylation served as positive control (Fig 4B).

Next, we tested OFD1 phosphorylation in cells stimulated with cAMP agonists. Cells were transiently transfected with OFD1 transgene, serum-deprived for 24 h and stimulated with forskolin (FSK), a diterpene that activates the adenylate cyclase and increases cAMP levels. Lysates were immunoblotted with anti-phospho-PKA substrate antibodies. As shown in Fig 4C and D, FSK treatment markedly increased phosphorylation of exogenous OFD1, by several fold over basal values. Similarly, endogenous OFD1 was phosphorylated in FSK-treated cells (Fig EV3A). To prove that S735 of the conserved PKA site was, indeed, the acceptor site for PKA transferred phosphate group, we generated a full length OFD1 mutant carrying S735 substituted to alanine (OFD1$_{S735A}$) and tested its phosphorylation in FSK-stimulated cells. Fig 4C and D shows that basal and FSK-induced phosphorylation of OFD1$_{S735A}$ mutant were dramatically downregulated, compared with wild-type protein. This finding indicates that S735 contributes to the overall cAMP-induced phosphorylation of OFD1. To demonstrate that PKA was, indeed, mediating the effects of FSK on OFD1 phosphorylation, FSK experiments were replicated in the presence of the PKA inhibitor H89. Fig 4C and D show that FSK-induced phosphorylation of OFD1 was almost completely abrogated by H89 treatment. Mass spectrometric analysis on affinity-isolated endogenous OFD1 from Hela cells transiently expressing PKAc-YFP confirmed that endogenous OFD1 was phosphorylated at S735 (Fig EV3B–D).

Since praja2, PKA and OFD1 are components of the complex assembled by TBC1D31 at centrosome, we hypothesized that phosphorylation of OFD1 by PKA might occur within the same multimeric complex. We tested this possibility by analysing FSK-induced OFD1 phosphorylation in cells devoid of praja2. Figure 4E and F shows that RNAi-mediated downregulation of praja2 markedly reduced FSK-induced phosphorylation of OFD1, supporting a model by which the phosphorylation of OFD1 by PKA occurs at centrosome within the TBC1D31-dependent multimeric complex (Fig 4G). We then tested the effects of S735 phosphorylation on TBC1D31/OFD1 interaction. As shown in Appendix Fig S4A and B, FSK treatment increased by about 50% the binding of OFD1 to TBC1D31. In contrast, the effect of FSK was almost abrogated by the S735A mutation (Appendix Fig S4A and B).

## PKA activation directs OFD1 to UPS

Since praja2 is an E3 ubiquitin ligase, we tested whether praja2 ubiquitylates OFD1. Figure 5A shows that overexpression of praja2

induced polyubiquitylation of OFD1. In contrast, the transfection of the inactive praja2 mutant (praja2rm) had no effect on OFD1 ubiquitylation. To further corroborate this notion, we tested whether praja2-dependent OFD1 ubiquitylation was induced by cAMP. FSK treatment effectively promoted OFD1 polyubiquitylation and, accordingly, praja2 silencing prevented FSK-induced ubiquitylation of OFD1 (Fig 5B).

Ubiquitylation is often coupled to proteolysis of target proteins (Kwon & Ciechanover, 2017). Therefore, we asked whether cAMP signalling activation induces OFD1 proteolysis. As shown Fig EV4A and B, the treatment of serum-deprived cells with cycloheximide, an antifungal antibiotic that inhibits protein synthesis, did not affect the bulk levels of OFD1. In contrast, FSK stimulation significantly reduced the half-life of OFD1 (Fig EV4A and B). The effects of FSK were replicated stimulating cells with prostaglandin E2 (PGE2), a ligand of the widely expressed G-protein coupled receptor EP2 and agonist of the cAMP•PKA pathway(Jiang & Dingledine, 2013; Jin *et al*, 2014). Thus, PGE2 treatment induced a time-dependent decline of OFD1 levels (Fig EV4C and D). Genetic silencing of praja2 prevented the decline of OFD1 levels in cAMP-stimulated cells (Fig 5C and D). Pre-treatment with MG132, a proteasome inhibitor, reversed downregulation of OFD1 by FSK stimulation (Fig EV4E and F). Next, we examined whether phosphorylation of OFD1 at S735 by PKA is required for FSK-induced ubiquitylation and proteolysis. Figure 5E shows that FSK-induced ubiquitylation of OFD1 was completely abrogated by the S735A mutant, compared with wild-type protein. Similarly, the levels of OFD1$_{S735A}$ mutant were unaffected by FSK treatment, over a time point stimulation (Fig 5F and G).

A wide variety of OFD1 mutations have been identified in sporadic and familiar forms of orofacial digital syndrome 1 (OFD1), affecting the synthesis, conformation or stability of the mutant protein (Macca & Franco, 2009). The most mutated residues (p.N75, p.H81, p.Y87, p.S92 and p.E97) are highly conserved among the species and mainly cluster within the Lis-H motif, a N-terminal domain likely involved in protein–protein interaction, protein folding and intracellular targeting. OFD1 E97G mutation is causally linked to a sporadic form of ciliopathy (Macca & Franco, 2009). We, thus, tested the relevance of this mutation in OFD1 stability. Figure 5H and I shows that the E97G mutation prevented the decline of OFD1 levels in the presence of FSK, suggesting a role of altered proteostasis for OFD1 in sporadic forms of ciliopathy.

## The TBC1D31-PKA-OFD1 axis controls ciliogenesis and signalling

Centrosomes are composed of two centrioles surrounded by a proteinaceous non-membrane-bound compartment, also called

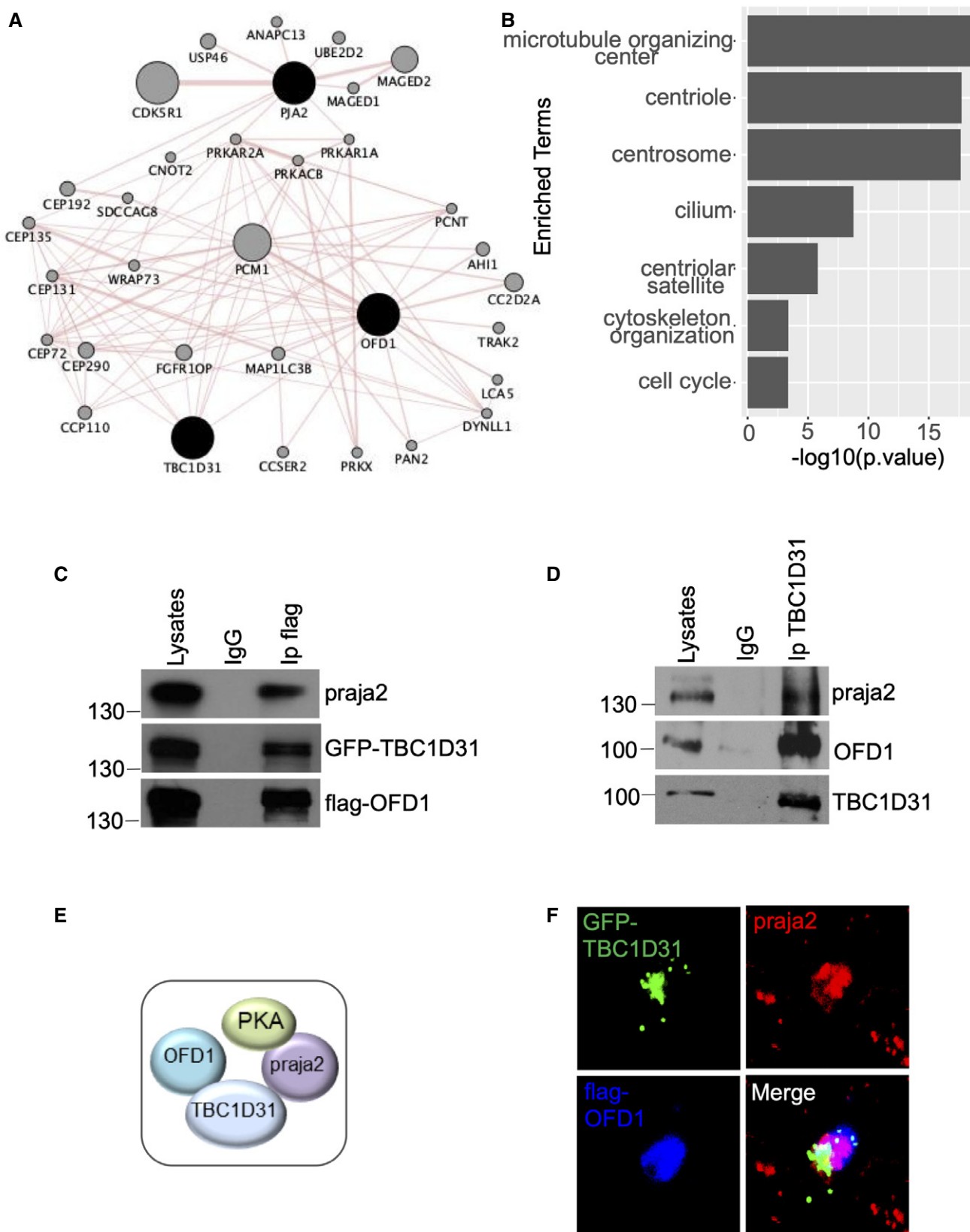

**Figure 3.**

◄

**Figure 3. OFD1 is a component of the TBC1D31/praja2 complex.**

A  Protein–protein interaction network of physical interactors of OFD1, praja2 and TBC1D31. Node size represents the degree of association of the proteins (grey nodes) associated with the input proteins set (black nodes). Edge size is representative of the strength of the mapped interaction between the two connecting nodes.

B  The bar plot reports the statistical significance of selected sets of Gene Ontology functional annotations enriched by the nodes in the network. Statistical significance is reported as -log10[*P*-value] on the X axis, enriched functional annotations are reported on the Y axis.

C  Lysates from HEK293 cells expressing GFP-TBC1D31 and flag-OFD1 were immunoprecipitated with anti-flag or control IgG. Endogenous praja2 and the overexpressed transgenes were revealed by immunoblot analysis.

D  Total lysates from mouse brain were immunoprecipitated with anti-TBC1D31 antibodies or with control IgG. The precipitates and an aliquot of lysates were immunoblotted with the indicated antibodies.

E  Schematic picture of the assembled TBC1D31/OFD1/praja2/PKA complex.

F  HEK293 cells were transiently co-transfected with GFP-TBC1D31 and flag-OFD1. Cells were fixed and stained for flag and praja2.

Source data are available online for this figure.

pericentriolar material (PCM). Centrosomes constitute the principal microtubule-organizing centre (MTOC) in mammalian cells and play a critical role in cell polarity, shape and spindle pole organization. In growth-arrested cells, centrioles form the basal bodies from where cilia elongate (Breslow & Holland, 2019). Given its localization at centrosome and pericentriolar satellites, we investigated the localization of TBC1D31 in ciliated cells. Cells were serum-deprived for 36 h to induce cilium formation, fixed and immunostained with antibodies directed against acetylated α-tubulin, a covalently modified form of tubulin that accumulates along the ciliary axoneme. Figure EV5A shows that TBC1D31 staining mostly localizes at the basal body of cilium. Next, we tested whether TBC1D31 was required for cilium biogenesis. We analysed primary cilia in serum-deprived HEK293 cells subjected to genetic silencing of TBC1D31. As shown in Fig 6A and B, downregulation of TBC1D31 severely decreased the number of ciliated cells compared with controls. These data were replicated in other cell types, such as glioblastoma cells (Fig EV5B and C).

Our data indicate that the cAMP-UPS axis controls OFD1 stability. Given the role of OFD1 in ciliogenesis, we speculated that regulation of OFD1 by PKA might impact on cilium biology. We tested this hypothesis by monitoring primary cilia in cells expressing either wild-type or the OFD1$_{S735A}$ mutant. Figure 6C (upper panels), Figs 6D and EV5D show that cilium morphology was significantly altered in OFD1$_{S735A}$ HEK293 cells, compared with control cells expressing wild-type OFD1. In OFD1$_{S735A}$ cells, the staining of acetylated tubulin appeared discontinuous, with narrowing and fragmentations along the entire length of cilium. Similar effects of the mutant protein were evident using a cilia-targeted GFP-reporter (cilia-APEX) (Fig 6C, middle panels). Cilium abnormalities induced by OFD1$_{S735A}$ mutant were also evident in mouse fibroblasts immunostained for ARL13B, a small GTPase of the ARF/ARL family that is localized along the ciliary membrane (Fig 6C, lower panels) (Caspary *et al*, 2007). The expression of either wild-type or mutant variant of OFD1 had no major impact on the number of ciliated cells, compared with controls (Fig 6E). Activation of the GPCR-cAMP pathway in serum-deprived cells induces cilium resorption (Lefebvre *et al*, 1978; Liang *et al*, 2016; Porpora *et al*, 2018). We therefore tested the impact of OFD1 phosphorylation on cAMP-induced cilium disassembly. Figure 6F and G shows that, in control cells and in cells expressing wild-type OFD1, FSK stimulation decreased the number of cilia. In contrast, in cells expressing the OFD1$_{S735A}$, FSK treatment had little impact on cilia number. These findings were confirmed in mouse fibroblasts NIH3T3 (Fig EV5E

and F). We conclude that PKA phosphorylation of OFD1 affects cilium abundance, morphology and dynamics.

To address the functional effects of mutant OFD1 on cilium-generated pathways, we analysed the Sonic Hedgehog (SHH) pathway in mouse fibroblasts transiently expressing either wild-type OFD1 or OFD1$_{S735A}$ mutant. The analysis was carried out in serum-deprived cells and in cells stimulated with purmorphamine, a small-agonist of the ciliary receptor *Smoothened*, that activates the downstream hedgehog pathway (Aanstad *et al*, 2009). As readout of SHH activity, we monitored the mRNA levels of the transcription factor *Gli2* (McCleary-Wheeler *et al*, 2020). Control cells (CMV) and wild-type OFD1 expressing cells show low levels of *Gli2* mRNA. Treatment with purmorphamine induced a marked accumulation of *Gli2* mRNA in both experimental groups. Figure 6H shows that purmorphamine-induced *Gli2* mRNA accumulation was completely abrogated by FSK treatment, in agreement with the inhibitory role of PKA activation on the SHH pathway (Chen *et al*, 2011; Vuolo *et al*, 2015). As suspected, expression of OFD1$_{S735A}$ mutant completely abrogated SHH-dependent accumulation of *Gli2* mRNA (Fig 6H).

**TBC1D31-PKA-OFD1 axis controls Medaka fish development**

To further prove the role of TBC1D31, which is conserved among vertebrates, in ciliogenesis, we carried out an in vivo analysis in the Medaka fish (Oryzias latipes, Ol) model system using gene knock-down, gene overexpression and rescue experiments. Morpholino (Mo) was designed against the medaka TBC1D31 ortholog present in the UCSC Genome Browser [(October 2005 v.1.0): ENSORLT00000007876.1 at chr16:11639389-11643923]. The efficiency and specificity of Mo (Gene Tools) were verified with the recommended controls (Eisen & Smith, 2008). From stage (St.) 24 onward, depletion of Ol-TBC1D31 caused a delay in embryonic development and evident embryonic morphological abnormalities, resulting in a significant body shape alteration associated to microcephaly, microphthalmia, pigmentation defects and pericardial oedema. Notably, morphological inspection did not reveal any apparent alteration in left-right asymmetry. These defects, which are similar to those observed in Nek10 morphants (Porpora *et al*, 2018), were progressive and led to a moderate embryonic lethality (~20%) (Fig 7A). Injection of morphants with human TBC1D31 mRNA rescued the whole phenotype, demonstrating that TBC1D31 function is critical for embryo development (Fig 7A). To determine whether the Ol-TBC1D31 knock-down (KD) phenotype was indeed related to abnormal ciliogenesis, we analysed cilia biogenesis on the apical

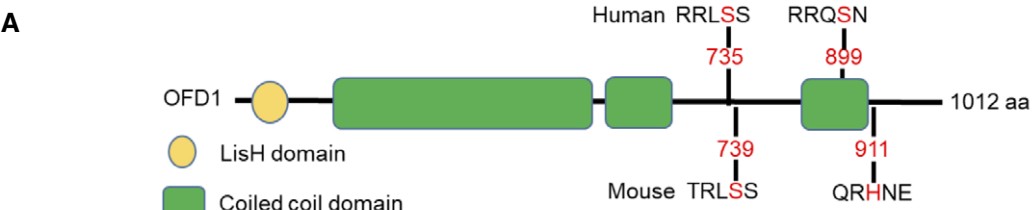

**A**

OFD1

○ LisH domain

▢ Coiled coil domain

Human RRL**S**S  RRQ**S**N
735  899

1012 aa

Mouse TRL**S**S  QR**H**NE
739  911

**B**

p(S/T)- PKA substrates

Ponceau S

**C**

**D**

**E**

**F**

**G**

Figure 4.

**Figure 4.  OFD1 is phosphorylated by PKA.**

A   Schematic representation of OFD1 domain organization, with the predicted PKA phosphorylation sites and the position of S735 and S899 in the human variant.
B   GST-fused, recombinant OFD1 polypeptides were used as substrates for *in vitro* phosphorylation assays using purified PKAc. GST-immobilized fusions were immunoblotted with an anti-phospho-(K/R)(K/R)X(S*/T*) antibody. A representative set of three independent experiments is shown. A Ponceau S staining of recombinant GST-OFD1 polypeptides is shown on the right panel. *Degradation products.
C   HEK293 cells transiently expressing flag-OFD1 or flag-S735A mutant were starved for 24 h and then treated with forskolin (FSK, 40 μM/1 h). Where indicated, cells were pre-treated with H89 (10 μM). Lysates were subjected to flag immunoprecipitation and immunoblot analysis with anti-phospho-PKA substrates and anti-flag antibodies. Anti-GFP IgG were used as control.
D   Quantitative analysis of the experiments shown in (C). A mean value ± SD of three independent experiments is shown. Student's *t* test, *$P < 0.05$, **$P < 0.01$.
E   Same as in (C), with the exception that cells were co-transfected with flag-OFD1 vector and with control siRNA (siCNT) or siRNA targeting praja2.
F   Quantitative analysis of the experiments shown in (E). A mean value ± SD of three independent experiments is shown. Student's *t* test, *$P < 0.05$.
G   Schematic picture of the assembled TBC1D31 complex showing that following cAMP stimulation PKA phosphorylates the co-assembled OFD1.

Source data are available online for this figure.

surface of cells of the neural tube at St.24–26 (2-days post-fertilization) of Medaka embryos development using whole-mount immunostaining with anti-acetylated α-tubulin. A significant reduction in cilia length was observed in large percentage of Mo-TBC1D31 morphant embryos (Fig 7B and C). Consistent with these observations, co-injection of human TBC1D31, which is not recognized by the morpholino Ol-TBC1D31, induced a statistically significant rescue of cilia length (Fig 7B and C). We, next, asked whether the Ol-TBC1D31 morphant phenotype was indeed related to abnormal phosphorylation/ubiquitylation of OFD1. We reasoned that if TBC1D31 directly controls ubiquitylation of OFD1 in vivo, overexpression of OFD1$_{S735A}$ mutant should induce similar phenotype because of alterations in ciliogenesis. Indeed, OFD1$_{S735A}$ injection caused embryonic morphological alterations accompanied by defects in ciliogenesis, culminating in cilia fragmentation (Fig 7A–C). Notably, ciliogenesis after overexpression of wild-type OFD1 was similar to wild-type embryos (Fig 7B and C). If most of the defects in ciliogenesis caused by TBC1D31 KD are due to abnormal phosphorylation/ubiquitylation of OFD1, co-injection of wild-type OFD1 should not restore normal ciliogenesis in TBC1D31 morphants. As predicted, wild-type OFD1 co-injection with the morpholino Ol-TBC1D31 did not rescue cilia length and Medaka embryogenesis (Fig 7A–C). To obtain additional support for the relevance of this TBC1D31-mediated regulation of OFD1, we also co-injected OFD1$_{S735A}$ with Mo-Ol-TBC1D31. Mo-Ol-TBC1D31/OFD1$_{S735A}$ injected embryos were morphologically indistinguishable from Mo-Ol-TBC1D31 morphants, showing similar defects in the ciliogenesis and embryo development, in which cilia resulted barely sketched (Fig 7A–C). If most of the changes in ciliogenesis caused by Ol-TBC1D31 KD are due to altered phosphorylation and proteolysis of OFD1 by praja2 activity, we reasoned that co-injection of OFD1$_{S735D}$ mutant, mimicking OFD1 phosphorylated form, combined with the dominant negative variant of human praja2 (hpraja2rm), should re-establish, at least in part, the ciliogenesis in Mo-Ol-TBC1D31 morphants and rescue larva phenotype. Consistent with the hypothesis, OFD1$_{S735D}$/hpraja2rm injection was sufficient to rescue the normal larva development (~20%) and partially recovered ciliogenesis defects in a substantial proportion of Mo-Ol-TBC1D31 morphants (Fig 7A–C). Importantly, injected larvae with OFD1$_{S735D}$ and Mo-Ol-TBC1D31 were morphologically indistinguishable from Mo-Ol-TBC1D31 morphants, further supporting the possibility that the alterations in ciliogenesis in Mo-Ol-TBC1D31 morphants are mediated by the altered phosphorylation/ubiquitylation of OFD1 (Fig 7A–C). The specificity of

this rescue was confirmed by additional controls described in Appendix Fig S5A–C.

Taken together, these data strongly support a primary role of TBC1D31 as a molecular scaffold located at the centrosome that controls the praja2/PKA/OFD1 molecular network necessary for the correct ciliogenesis and Medaka fish development (Fig 7D).

## Discussion

Here, we report the identification of TBC1D31 as a molecular scaffold located at the centrosome that plays a major regulatory role in cilium biogenesis. We identified OFD1, praja2 and PKA as components of the TBC1D31-assembled complex and characterized the functional interplay between these elements in critical aspects of ciliogenesis. Within this regulatory circuitry, GPCR activation induced PKA phosphorylation, ubiquitylation and proteolysis of OFD1. By modulating OFD1 turnover, the TBC1D31 complex controls important events underlying ciliogenesis and development.

TBC1D31 is a highly conserved protein originally identified by proteomic screening as component of the ciliary network in mammalian cells. TBC1D31 localizes at the centrosome and pericentriolar satellites (Gupta *et al*, 2015; Gheiratmand *et al*, 2019). However, the biological functions and the role of TBC1D31 in cilium biology were largely unknown. We identified TBC1D31 as binding partner of praja2. A combinatorial approach of *in vitro* binding assays, deletion mutagenesis, genetic knock-down and *in situ* immunostaining analysis disclosed a major role for TBC1D31 in mediating the targeting of praja2 to centrosome and centriolar satellites. A minimal core domain of praja2 was identified as essential in mediating the interaction with the extreme C-terminus of TBC1D31. 3D-modelling prediction, structural analyses and *in vitro* binding assays helped reveal the molecular basis of TBC1D31/praja2 interaction, identifying the relevant residues involved in the interaction. In particular, the TBC1D31 941–970 region contains many positively charged residues and we identified R957, R959 and H960 as the key residues involved in the interaction. Functional analysis demonstrated that TBC1D31 has an essential role in primary ciliogenesis. Moreover, downregulation of TBC1D31 in Medaka fish impaired cilium elongation and severely affected fish development. These findings point to a highly conserved role of TBC1D31 in coordinating signalling pathways and the ubiquitin system at centrosome and centriolar satellites, with important implications in ciliogenesis.

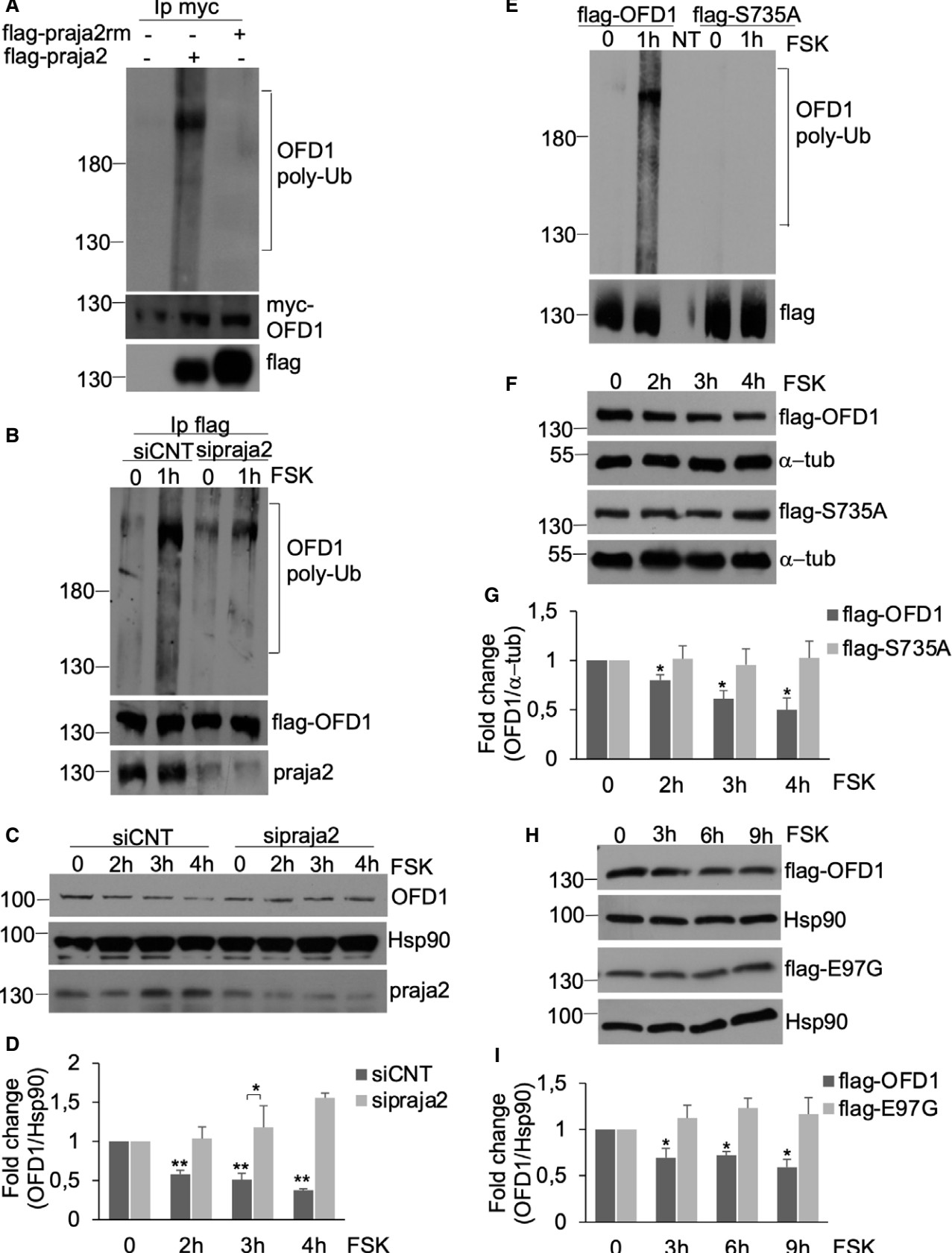

**Figure 5.**

**Figure 5. cAMP targets OFD1 to UPS.**

A   Immunoprecipitation of OFD1 from HEK293 cell lysates expressing myc-OFD1, ubiquitin-HA and flag-praja2 (or flag-praja2rm). The precipitates were immunoblotted with anti-HA (ubiquitinated OFD1) and anti-myc antibodies. Flag-praja2 expression was evaluated in total lysates.

B   Flag-OFD1 and ubiquitin-HA vectors were co-transfected with control siRNA or siRNA targeting praja2 (sipraja2). Serum-deprived cells were left untreated or stimulated with FSK for 1 h. Ubiquitinated OFD1 and the levels of endogenous praja2 were detected as in (A).

C   HEK293 cells transfected with siRNAs were serum-deprived, pre-treated with cycloheximide and stimulated with FSK at different time points. Lysates were immunoblotted for endogenous praja2 and OFD1. Hsp90 was used as loading control.

D   Quantitative analysis of the experiments shown in (C). A mean value $\pm$ SD of three independent experiments is shown. Student's $t$ test * $P < 0.05$; ** $< 0.01$ versus sipraja2 (+FSK) and basal values (0).

E   Ubiquitination of flag-OFD1 or flag-S735A in HEK293 cells treated with FSK for 1 h.

F   Immunoblot analysis of flag-OFD1 or flag-S735A in HEK293 cells stimulated with FSK as in (C). The levels of $\alpha$-tubulin were used as loading control.

G   Quantitative analysis of the experiments shown in (F). A mean value $\pm$ SD of three independent experiments is shown. Student's $t$ test *$P < 0.05$ versus S735A mutant (3 h and 4 h, +FSK) and basal values (0).

H   Immunoblot analysis of flag-OFD1 or flag-E97G mutant in HEK293 cells stimulated with FSK as in (C).

I   Quantitative analysis of the experiments shown in (H). A mean value $\pm$ SD of three independent experiments is shown. Student's $t$ test, *$P < 0.05$ versus E97G mutant (+FSK) and basal values (0).

Source data are available online for this figure.

---

At mechanistic level, OFD1 was identified as a molecular target of the regulatory circuitry governed by TBC1D31. We found that PKA directly phosphorylates OFD1 at serine 735. Phosphorylation of OFD1 was largely abrogated by praja2 silencing, supporting the role of praja2 as a PKA-anchor protein. This is especially relevant for hormone-stimulated cells where praja2 links phosphorylation to ubiquitylation of colocalized protein kinases, scaffolds and effectors, impacting on differentiation, cell proliferation and tumour growth (Rinaldi *et al*, 2015). Several AKAP members have been identified as regulators of cAMP signalling at centrosome/centriolar satellites (Diviani *et al*, 2000; Zhong *et al*, 2009; Kolobova, Roland *et al*, 2017; Bucko *et al*, 2019). Here, we show that a pool of PKA bound to praja2 controls OFD1 phosphorylation. This finding adds further insight into the complexity of ciliary cAMP signalling, suggesting the existence of a regulatory mechanism by which PKA acts on substrate(s) assembled within the same transduction unit. Within this molecular platform, praja2 ligates ubiquitin moieties to selected PKA targets. Thus, in response to cAMP stimulation, praja2 efficiently couples phosphorylation to ubiquitin-dependent proteolysis of OFD1. By removing OFD1, praja2 downregulates ciliogenesis and signalling. This represents the first evidence of the involvement of the ubiquitin-proteasome pathway in the control of OFD1 stability.

The onset of ciliogenesis is marked by selective removal of a pool of OFD1 from the pericentriolar satellites through the autophagy machinery. Interfering with this autophagic circuitry reduced the number and length of primary cilia (Tang *et al*, 2013). Mechanistically, the loss of OFD1 induces the translocation of BBS4 (Bardet-Biedl syndrome 4) to the ciliary compartment. BBS4 is an essential component of the BBSome, an octameric protein complex involved in ciliogenesis and ciliary trafficking. Mutations of BBS subunits have been causally linked to severe human syndromes (Bardet-Biedl syndrome) characterized by obesity, renal abnormalities, polydactyly, retinal dystrophy and other developmental defects. However, whether autophagy promotes or inhibits ciliogenesis is still unresolved (Lam *et al*, 2013; Pampliega *et al*, 2013). On the other hand, OFD1 is essential for ciliogenesis since it controls centriole distal appendage formation and the recruitment of intraflagellar transport protein Ift88 within the cilium. By stabilizing the length of centriolar microtubules, OFD1 supports cilium biogenesis and elongation. This was confirmed by the identification of disease-linked inactivating mutations of OFD1 that impair the assembly of centriolar microtubules, centriolar elongation and ciliogenesis (Singla *et al*, 2010). These findings imply that OFD1 works at multiple levels in the ciliogenesis programme, since mutations of OFD1 may affect localization and stability of mutant protein, and/or its interaction with relevant ciliary partners. As consequence, cilium onset, axonemal elongation, ciliary trafficking and downstream signalling are deregulated in cells harbouring OFD1 mutations.

The cAMP cascade plays a relevant role in cilium biology. Components of the cAMP generating system, effectors and regulators

---

**Figure 6. The TBC1D31 signalling circuitry controls ciliogenesis.**

A   siRNA transfected HEK293 cells were serum-deprived and stained for acetylated-tubulin, TBC1D31 and DRAQ5.

B   Quantitative analysis of the experiments shown in (A). A mean value $\pm$ SD of four independent experiments is shown. Student's $t$ test *$P < 0.05$.

C   HEK293 cells expressing flag-OFD1 or flag-S735A were serum-deprived and subjected to staining analysis for acetylated $\alpha$-tubulin, flag and DRAQ5. Images were collected by super-resolution microscopy (upper panels). Where indicated, flag-OFD1 or flag-S735A was co-transfected with cilia-APEX-GFP vector (middle panels). NIH3T3 expressing flag-OFD1 or flag-S735A were stained with anti-ARL13B and anti-flag antibodies.

D, E   Statistical analysis of the experiments shown in (C). A mean value $\pm$ SD of three independent experiments is shown. For each experimental group, a minimum of 40 cilia were analysed. Student's $t$ test ***$P < 0.001$.

F   HEK293 cells transiently transfected with flag-OFD1 or flag-S735A were serum-deprived, treated with FSK (6 h) and stained for flag, acetylated-tubulin and DRAQ5.

G   Statistical analysis of the experiments shown in (F). A mean value $\pm$ SD of three independent experiments is shown. Cells analysed for each experiment: 300 for NT, 80 for flag-OFD1 and 80 for flag-S735A. Student's $t$ test, *$P < 0.05$, **$P < 0.01$.

H   Quantitative RT–PCR analysis showing levels of Gli2 mRNA in NIH3T3 fibroblasts transfected with vectors for flag-OFD1 and flag-S735A. An empty vector (CMV) was used as control. Serum-deprived cells were stimulated with the SHH ligand purmorphamine (1 $\mu$M) for 24 h. Where indicated, cells were treated with FSK for 6 h before harvesting. The data represent a mean value $\pm$ SD of three independent experiments. Student's $t$ test, $P$** $< 0.01$.

Source data are available online for this figure.

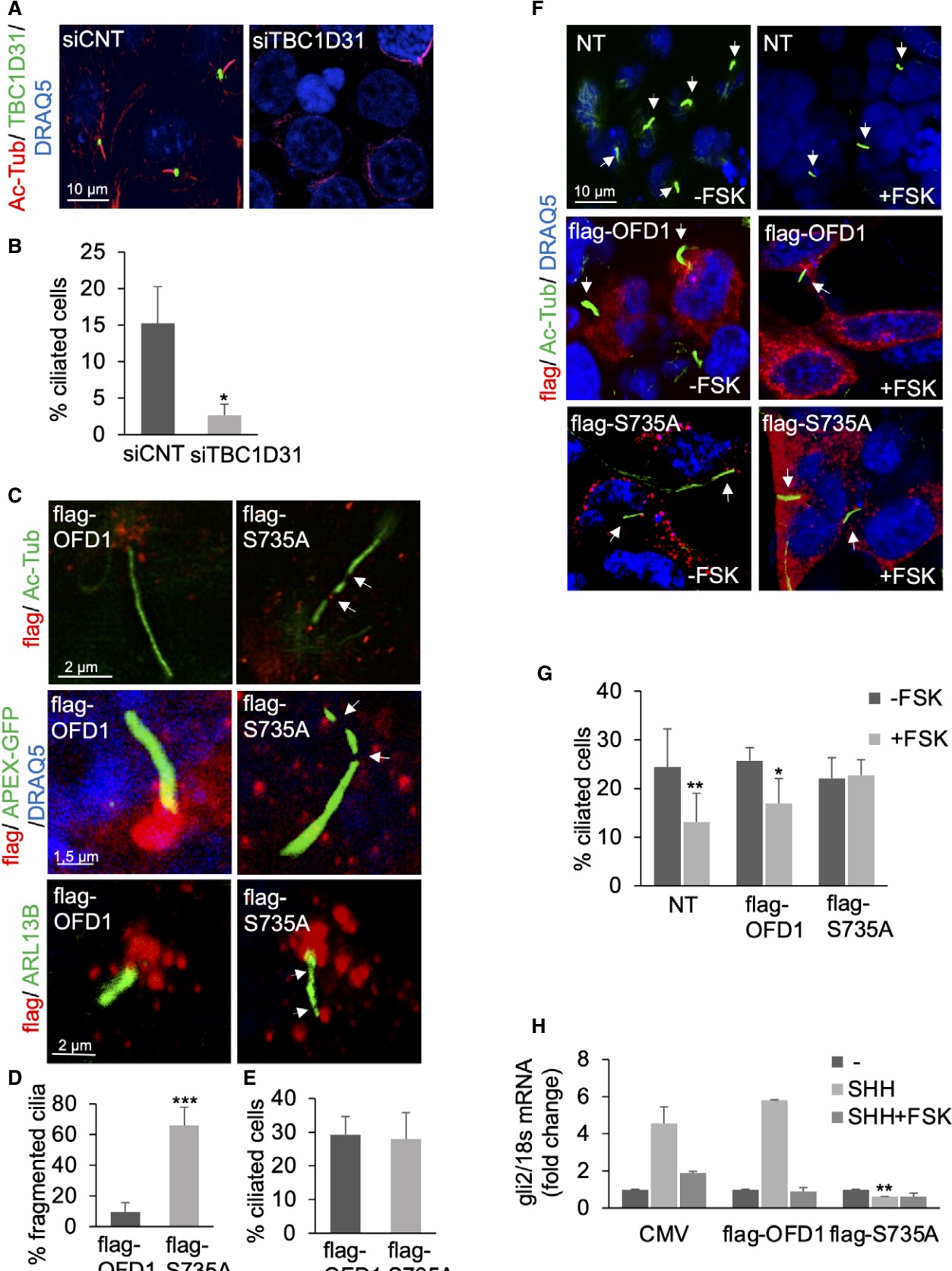

Figure 6.

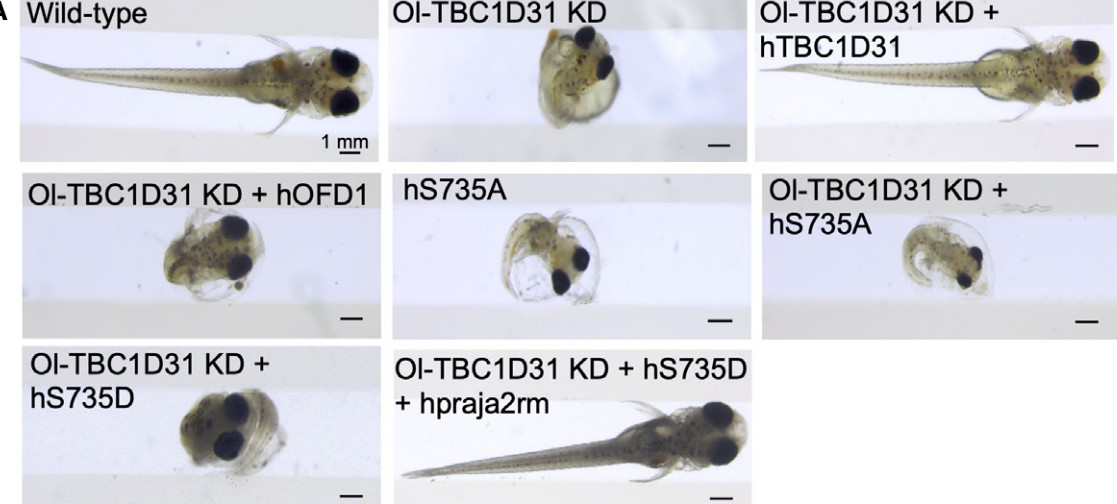

Figure 7.

◄

**Figure 7. TBC1D1 and PKA/OFD1 pathway controls Medaka fish development.**

A  Stereo-microscopic images of wild-type, Ol-TBC1D31 KD, Ol-TBC1D31 KD + hTBC1D31, Ol-TBC1D31 KD + wild-type hOFD1, hOFD1$_{S735A}$, Ol-TBC1D31 KD + hOFD1$_{S735A}$, Ol-TBC1D31 KD + hOFD1$_{S735D}$ and Ol-TBC1D31 KD + hOFD1$_{S735D}$ + hpraja2rm injected Medaka larvae, at stage 40.
B  Confocal images of cilia of the neural tube cells in the wild-type, Ol-TBC1D31 KD, Ol-TBC1D31 KD + hTBC1D31, wild-type hOFD1, Ol-TBC1D31 KD + wild-type hOFD1, hOFD1$_{S735A}$, Ol-TBC1D31 KD + hOFD1$_{S735A}$, Ol-TBC1D31 KD + hOFD1$_{S735D}$ and Ol-TBC1D31 KD + hOFD1$_{S735D}$ + hpraja2rm stained with anti-acetylated α-tubulin antibody (green) and DAPI (blue).
C  In the graph is reported the cilia length in wild-type, Ol-TBC1D31 KD, Ol-TBC1D31 KD + hTBC1D31, wild-type hOFD1, Ol-TBC1D31 KD + wild-type hOFD1, hOFD1$_{S735A}$, Ol-TBC1D31 KD + hOFD1$_{S735A}$, Ol-TBC1D31 KD + hOFD1$_{S735D}$ and Ol-TBC1D31 KD + hOFD1$_{S735D}$ + hpraja2rm. The data are expressed as mean value ± SE of twelve independent experiments. Student's *t* test, ***$P \leq 0.001$.
D  TBC1D31 assembled complex at centrosome positively regulates ciliogenesis and signalling. Ligand (L)-induced stimulation of G-protein coupled receptors (GPCRs) activates the adenylate cyclase (AC) and elevates cAMP levels, which in turn activates PKA. PKA phosphorylates OFD1 at S735, priming the ciliopathy protein to praja2-dependent ubiquitylation and proteasomal degradation. Decreasing the levels of OFD1 negatively impacts on cilium biogenesis and downstream signalling.

have been identified within the ciliary compartment. Within the cilium, the mutual regulation between PKA and Hedgehog pathway finely controls nuclear gene transcription (Vuolo *et al*, 2015; Moore *et al*, 2016; Mukherjee *et al*, 2016; Siljee *et al*, 2018; Sherpa *et al*, 2020). cAMP also controls cilium dynamics. Thus, in serum-deprived mammalian cells, activation of cAMP cascade promotes cilium resorption (Porpora *et al*, 2018). This regulatory mechanism is highly conserved in flagellated unicellular eukaryotes (Lefebvre *et al*, 1978; Liang *et al*, 2016). The mechanism of cAMP action on cilium stability relies, in part, on the PKA-induced ubiquitylation and proteolysis of the pro-ciliogenic Nima-related kinase 10, NEK10 (Porpora *et al*, 2018). cAMP•PKA signalling can also support cilium elongation. Thus, in serum supplemented confluent cells, cAMP signalling increased the cilium length (Besschetnova *et al*, 2010; Porpora *et al*, 2018). The different growth conditions may, thus, shift the sensitivity of the system to cAMP and/or modulate the dissemination of cAMP signals to the ciliary compartment. In this context, nanobody-based optogenetic tools have revealed the spatial contribution of cAMP signalling in cilium biology and dynamics. Thus, high cAMP levels within the cilium increase the axonemal length, whereas accumulation of cAMP within the cell body by membrane GPCR activation reduced cilium length (Hansen *et al*, 2020).

The UPS plays a relevant role for primary ciliogenesis (Shearer & Saunders, 2016). Accordingly, we found that inhibition of the proteasome significantly impacted on ciliated cells (Appendix Fig S6A and B). We show above that, in response to cAMP stimulation, the praja2-UPS axis reduced the bulk levels of OFD1, decreasing the number of ciliated cells. The effects of cAMP-UPS cascade on ciliated cells were mostly abolished by expressing a phospho-deficient OFD1 mutant. Moreover, marked alterations of cilium morphology were evident after expression of an OFD1 phospho-mutant. A defect in cilium morphology was coupled to a dramatic impairment of SHH-dependent transcription of *Gli2* mRNA. This ciliary phenotype was recapitulated in Medaka fish where knock-down of TBC1D31 markedly altered ciliogenesis, cilium morphology and development. Importantly, this phenotype might be a consequence of lack of OFD1, because previous studies have shown that depletion of OFD1 directly induces this phenotype in zebrafish (Ferrante *et al*, 2009; Liu *et al*, 2014). However, we found that the concomitant knock-down of TBC1D31 and overexpression of wild-type OFD1 was insufficient to restore normal cilia size and embryogenesis (Fig 7A–C), possibly because TBC1D31 depletion directly affects praja2/PKA/OFD1 molecular network at centrosome. Consistent with this hypothesis, overexpression of OFD1$_{S735A}$ correlated with a significant alteration of ciliogenesis that was associated with abnormal

embryogenesis. Concordantly, co-injections of Mo-Ol-TBC1D31/OFD1$_{S735A}$ did not exacerbate the morphant phenotype. Moreover, OFD1$_{S735D}$/hpraja2rm injection was sufficient to rescue ciliogenesis defects and normal larva development supporting the model by which TBC1D31 controls the praja2/PKA/OFD1 functional complex.

The N-terminal LisH domain of OFD1 is highly conserved in multiple proteins. It participates in a variety of activities, including protein–protein interaction, protein stability and intracellular localization. Missense mutations within the LisH domain of OFD1, including the E97G substitution, are causally linked to sporadic forms of ciliopathy (Macca & Franco, 2009). We found that the E97G mutation alters the proteolytic turnover of OFD1 in cells stimulated with cAMP, suggesting a pathogenic role of altered proteostasis of OFD1 in certain forms of ciliopathy disorders.

In summary, we have identified a centrosomal transduction hub that finely controls the timing and diffusion of cAMP signalling to ciliary targets. In response to GPCR activation, the TBC1D31-driven signalling system dynamically couples PKA phosphorylation of OFD1 to its ubiquitylation and proteolysis. By controlling OFD1 levels, this regulatory circuitry deeply impacts on cilium biology and development, with important implication for ciliopathies and proliferative disorders.

# Materials and Methods

### Cell lines and tissues

Human embryonic kidney cell line (HEK293), mouse fibroblasts (NIH3T3) and human glioblastoma cells (U-87 MG) were cultured in Dulbecco modified Eagle's medium containing 10% foetal bovine serum (FBS) in an atmosphere of 5% $CO_2$. Hela Kyoto cells were cultured with EMEM 10% (FBS).

### Plasmids and transfection

Vector encoding for TBC1D31-GFP were provided by Dr Laurence Pelletier; flag-OFD1 and OFD1-Myc, praja2 and HA-ubiquitin were previously described (Giorgio *et al*, 2007; Lignitto *et al*, 2011a); A praja2 inactive mutant (praja2rm) carrying cys634 and cys637 changed to alanine was used; E97G-flag was purchased from Genscript; pEF5B-FRT-cilia-APEX (APEX-GFP) was purchased from Addgene; TBC1D31 mutant (940–970) and OFD1 mutants were generated by PCR using specific oligonucleotides. siRNAs targeting praja2 and TBC1D31 were purchased from Dharmacon and Life

technologies, respectively. siRNAs were transfected using Lipofectamine 2000 (Invitrogen) at a final concentration of 100 pmol/ml of culture medium. The siRNA sequences of TBC1D31 (Life Technologies) are: sequence 1: 5'- GGAAUGCACUUACAAGAUTT- 3'; sequence 2: 5'- GCUACUCAGAGUAUUACAGTT- 3'; The siRNA sequences of praja2 (Dharmacon) are as follows: sequence 1: 5'- GAGAUGAGUUUGAAGAGUU- 3'; sequence 2: 5'- GGGAGAAAUUC- CUUGGUUA- 3'; sequence 3: 5'- UGACAAAGAUGAAGAUAGU- 3'; sequence 4: 5'- UCAGAUGACCUCUUAAUAA- 3'. Control siRNA was purchased from Ambion (am4637).

### Antibodies and chemicals

Primary antibodies against the following epitopes were used: GFP (1:1,000 immunoblot; 1:50 immunoprecipitation; #11814460001, Roche), flag (1:5,000 immunoblot, 1:200 immunoprecipitation; 1:1,000 immunofluorescence; #F3165, Sigma-Aldrich), GST (1:5,000; #sc-138 Santa Cruz Biotechnology), praja2 (1:1,000 immunoblot, 1:200 immunofluorescence; #A302-991A, Bethyl Laboratories), γ-tubulin (1:300; #sc-17788, Santa Cruz Biotechnology), TBC1D31 (1:100 immunofluorescence; 1:500 immunoblot; #ab121771, Abcam), TBC1D31 (1:1,000 immunoblot, 1:100 immuno-precipitation, 1:100 immunofluorescence; custom made, Primm srl, Italy), HA.11 (1:1,000; #16B12, BioLegend), OFD1 (1:1,000 immunoblot; 1:00 immunoprecipitation; #ABC961, Sigma-Aldrich), OFD1 (1:100 immunofluorescence; #HPA031103, Sigma-Aldrich), phospho-(K/R) (K/R)X(S*/T*) antibody (1:1,000; #9621S, Cell Signalling Technology), mouse acetylated -tubulin (1:400; #T7451, Sigma-Aldrich), rabbit acetylated-tubulin (1:400; #ab125356, Abcam), myc (1:1,000; #M4439, Sigma-Aldrich), tubulin (1:5,000; #T6199, Sigma-Aldrich), Hsp90 (1:5,000; #60318-1-Ig, Cell Signalling Technology), Hsp70 (1:5,000; #10995-1-AP, Cell Signalling Technology), PKAc (1:1,000; sc-28315, Santa Cruz), β-actin (1:700; #A2228, Sigma-Aldrich), ARL13B (1:500; #17711-I-AP, Proteintech). Antibody-antigen complexes were detected by HRP-conjugated antibodies (Bio-Rad Laboratories) and ECL (EuroClone). The following chemicals were used: forskolin (40 μM; #F3917, Sigma-Aldrich), H89 dihydrochloride hydrate (10 μM; #B1427, Sigma-Aldrich), prostaglandin E2 (1 μM; #P0409, Sigma-Aldrich), cycloheximide (100 μM; #C104450, Sigma-Aldrich), MG132 (10 μM; #C2211, Sigma-Aldrich); Purmorphamine (1 μM; # ab120933 Abcam); bafilomycin (1 μM; #B1793, Sigma-Aldrich).

### Immunoprecipitation, pull-down assays and immunoblot

Cells were lysed with buffer containing 0.5% NP40 (150 mM NaCl, 50 mM Tris–HCl pH 7.5, EDTA 1 mM and 0.5% NP40) and supplemented with protease inhibitors, phenylmethylsulfonyl fluoride (PMSF) and phosphatase inhibitors. For ubiquitylation assays, cells were lysed with triple detergent buffer (150 mM NaCl, 1% NP40, 0.1% SDS, 50 mM Tris–HCl pH 8, 0.5% NaDOC). The lysates were incubated overnight with the indicated antibodies for the immunoprecipitation. For pull-down assay, lysates were incubated three hours with GST-fused proteins immobilized on glutathione beads. In both cases, pellets were washed three times with lysis buffer. Medaka embryos at St.24 were lysed with buffer containing 10 mM Tris–HCl pH 8.0, 0.2% SDS and supplemented with protease inhibitors, PMSF, phosphatase inhibitors and sodium orthovanadate.

Precipitates and 50 μg of whole cell and embryos lysates were loaded on SDS polyacrylamide gel and transferred on nitrocellulose membrane. Filter was blocked with 5% milk in TBS Tween 0.1%, incubated with primary and secondary antibodies and finally proteins were detected with ECL (EuroClone). For mass spectrometric analysis, Hela Kyoto cells growing in EMEM supplemented with 5 mM L-Glutamine, 2 mM Pyruvate and 10%FBS were seeded in 15cm-dishes and after 24 h were transfected with the PKAc-YFP plasmid (Eccles *et al*, 2016) and serum-deprived for 48 h. As additional experimental groups, we included serum-deprived control cells and cells stimulated with FSK for 1 h. Then, the cells were washed with PBS and lysed with standard lysis buffer (10 mM sodium phosphate (pH 7.2), 150 mM NaCl, 0.5% Triton X-100 supplemented with standard protease inhibitors and phosphatase inhibitors. After lysate clarification (13,000 rpm, 20 min) and performed IPs using Protein A/G mixtures and 2 μg of control (HA antibody, Cell Signaling) or anti-OFD1 (Sigma-Aldrich) for 16 h at 4°C. Resin-associated proteins were washed six times with standard lysis buffer, eluted with Laemmli sample buffer and separated by SDS–PAGE. OFD1-containing gel fragments from the three experimental groups were isolated and subjected to mass spectrometric analysis.

### Mass spectrometric analysis

In-gel digestion was performed as described (Shevchenko *et al*, 2006). Briefly, shrinking and swelling was achieved with pure acetonitrile (ACN) and 100 mM NH4HCO3 (ABC). For reduction of disulphide bonds, 10 mM dithiothreitol (dissolved in 100 mM ABC) was used and alkylation of cysteine residues was achieved with 55 mM iodacetamide (dissolved in 100 mM ABC). For trypsin digestion, the gel pieces were covered with a trypsin solution (6.5 ng/μL sequencing-grade trypsin, dissolved in 10 mM ABC containing 10% ACN) and incubated at 37°C for 16 h. Tryptic peptides were extracted with 5% FA, 50% ACN and evaporated. For further LC-MS/MS analysis, samples were dissolved in 15 μL 0.1% FA. For LC-MS/MS analysis, 6 μl of the tryptic peptide digest was injected on a nano-ultra pressure liquid chromatography system (Dionex UltiMate 3000 RSLCnano pro flow, Thermo Scientific, Bremen, Germany) coupled via electrospray-ionization (ESI) source to a tribrid orbitrap mass spectrometer (Orbitrap Fusion Lumos, Thermo Scientific, San Jose, CA, USA). The samples were loaded (15 μL/min) on a trapping column (nanoE MZ Sym C18, 5 μm, 180 μm × 20 mm, Waters, Germany, buffer A: 0.1% FA in HPLC-H2O; buffer B: 80% ACN, 0.1% FA in HPLC-H2O) with 5% buffer B. After sample loading, the trapping column was washed for 2 min with 5% buffer B (15 μL/min) and the peptides were eluted (250 nL/min) onto the separation column (nanoE MZ PST CSH, 130 A, C18 1.7 μ, 75 μm × 250 mm, Waters, Germany) and separated with a gradient of 5–30% B in 25 min, followed by 30–50% in 5 min. The spray was generated from a steel emitter (Thermo Fisher Scientific, Dreieich, Germany) at a capillary voltage of 1900 V. For MS/MS analysis, a targeted MS/MS method was used in positive ion mode. The mass selective quadrupole was set to m/z 426.5708, which corresponds to the monoisotopic mass of the triply charged OFD1 phosphopeptide LpSSTPLPKAKR ([M + 3H]3+), with isolation window of 0.4 Da centred along the monoisotopic m/z value. A normalized HCD collision energy of 30% was used, the fragment ions were recorded in the orbitrap with a resolution of 240,000 at

m/z 200 using an AGC target of 200% and maximum injection time of 502 ms. Next to the targeted MS/MS method, MS/MS spectra were recorded in data-dependent acquisition (DDA) mode using top-speed mode. Full spectra were recorded in the orbitrap with a resolution of 60,000 at *m/z* 200 over a m/z-range from 350–1,600 *m/z* using the ETD source for internal mass calibration, an AGC target of 100% and a maximum injection time of 50 ms. MS/MS spectra were recorded in top-speed mode using a normalized HCD collision energy of 30% and an isolation window of 0.4 Da. MS/MS spectra were recorded in the orbitrap with a resolution of 15,000 (*m/z* at 200), an AGC target of 100% and a maximum injection time of 100 ms. For data analysis and visualization, FreeStyle 1.6 (Version 1.6.75.20, Thermo Scientific, Bremen, Germany). Extracted ion chromatograms (EIC) for the fragment ions of the phosphopeptide LpSSTPLPKAKR were generated with a mass tolerance of ± 3 ppm. For peptides and protein identification, LC-MS raw data were processed with Proteome Discoverer 2.4 (Thermo Scientific, Bremen, Germany). For identification, MS/MS spectra were searched with Sequest HT against the human SwissProt database (www.uniprot.org, downloaded August 16, 2019, 20,368 entries) and a contaminant database (116 entries). The searches were performed using the following parameters: precursor mass tolerance was set to 10 ppm and fragment mass tolerance was set to 0.02 Da. Furthermore, two missed cleavages were allowed and a carbamidomethylation on cysteine residues as a fixed modification. In addition, an oxidation of methionine and a phosphorylation at serine was considered as variable modifications. Peptides and proteins were identified with a FDR of 1% using Percolator. Phosphorylation of OFD1-S735 was identified in all experimental groups.

### Immunofluorescence and confocal analysis

Cells were plated on poly-D-lysine (Sigma) coated glass. Then cells were fixed with paraformaldehyde 5%, permeabilized with PBS 0.3% Triton, blocked with PBS 5% Bovine Serum Albumin (SERVA) and immunostained with the indicated primary antibody. Signals were revealed using fluorescent- or rhodamine conjugated secondary antibodies (1:200; Invitrogen). Nuclei were stained with the cell permeable DNA fluorescent dye DRAQ5™ (Abcam ab108410). For cilia counting, we considered as cut-off for cilia the length ≥ 1 μm. Immunostaining was visualized using a Zeiss LSM 510 META laser scanning confocal microscope. Where indicated, high-resolution images were acquired with a Zeiss LSM 880 confocal microscope equipped with a Plan-Apochromat 63x/1.4 oil immersion objective. Where indicated, high-resolution images were acquired with a Zeiss LSM 880 confocal microscope equipped with Airyscan super-resolution imaging module, using a Å ~ 63/1.40 NA Plan-Apochromat Oil DIC M27 objective lens (Zeiss MicroImaging, Jena, Germany).

### PKA *in vitro* phosphorylation assay

Fusion proteins carrying distinct segments of OFD1 fused to the C-terminus of GST were previously described (Giorgio *et al*, 2007). GST-OFD1 polypeptides were expressed in the E. Coli strain BL21-DE3-RIL and expression was induced with 0.8 mM isopropyl-β-D-thiogalactoside (IPTG) at 16°C/16 h. Cells were collected by centrifugation, resuspended in PBS/0.5% Triton and lysed at 1,300

psi using a French press device. Clarified lysates were subjected to GST purifications using glutathione-sepharose beads (GE Healthcare) following the supplier's instructions. His6-PKAc was expressed and purified as previously described (Porpora *et al*, 2018). For the phosphorylation reaction, equal amounts of GST-OFD1 protein beads were incubated with recombinant His6-PKAc in phosphorylation buffer (40 mM Tris at pH 7.5, 0.1 mM EGTA, 10 mM ATP and 10 mM MgCl$_2$) at 30°C/20 min at 1,000 rpm. Beads were washed four times with PBS/0.5% Triton, subjected to SDS–PAGE and immunoblotting using an anti-phospho-(K/R)(K/R)X(S*/T*) specific antibody.

### Microscale thermophoresis (MST)

MST experiments were performed on a Monolith NT 115 system (Nano Temper Technologies) using 100 % LED and 20% IR-laser power. Recombinant GST-praja2$_{530-570}$ (P1), GST-praja2$_{550-610}$ (P2) and GST-praja2$_{590-630}$ (P3) proteins were purified by a two-step purification procedure consisting of a GST Trap (GE-Healthcare, Milan, Italy) and a gel-filtration chromatography (Superdex 75 10 × 300, GE-Healthcare, Milan, Italy). Peptides of the C-terminus segment of TBC1D31 (WDTTGQNLIKKVRNLRQRLTARARHRCQTPHLLAA), double AA mutant TBC1D31 (WDTTGQNLIKKV<u>A</u>NL<u>A</u>QRLTARARHRCQTPHLLAA) and triple ADA mutant TBC1D31 (WDTTGQNLIKKVRNLRQRLTA<u>A</u>AD<u>A</u>RCQTPHLLAA) were purchased from Shanghai Apeptide Co., Ltd (Pudong District, Shanghai, China). To get a precise quantification, peptide sequences were extended up to the W937. Recombinant proteins were labelled with reactive dyes using N-Hydroxysuccinimide (NHS)-ester chemistry, which reacts efficiently with the primary amines of the proteins to form a stable dye-protein conjugate. For labelling, protein concentration was adjusted to 20 μM in labelling buffer (Nano Temper Technologies), while the dye concentration was adjusted to a threefold concentration of the protein (60 μM). The protein and the fluorescent dye solution were incubated for 60 min at room temperature in the dark. A 16-point serial dilution (1:1) was prepared for WA35 TBC1D31 at the final concentration ranged from 500 μM to 15 nM in the experiments with GST-praja2$_{530-570}$ and GST-praja2$_{590-630}$, and 450 μM to 13 nM with GST-praja2$_{550-610}$. The samples were filled into Premium capillaries. Measurements were conducted at 25°C in 20 mM NaP, 200 mM NaCl, 1 mM DTT, 0.05 % Tween-20 pH 7.4 buffer. As negative control, the same experiments were performed in the presence of GST-labelled protein alone. An equation implemented by the software MO-S002 MO Affinity Analysis provided by the manufacturer was used for fitting baseline-corrected normalized fluorescence (ΔFNorm%) values at different peptides concentrations. The experiments in the same experimental conditions were carried out also between P1 and TBC1D31$_{AA}$ (650 nM-19 nM) or TBC1D31$_{ADA}$ (550 nM-16 nM).

### Circular dichroism (CD)

CD spectra were recorded at 20°C on a Jasco J-810 spectropolarimeter (JASCO Corp, Milan, Italy). Far-UV measurements (190–260 nm) were carried out using a 0.1-cm path length cell in 10 mM sodium phosphate pH 7.4. The spectra of the wild-type TBC1D31, TBC1D31$^{AA}$ and TBC1D31$_{ADA}$ peptides were performed at a concentration of 40 μM. Spectra derived from the average of three scans,

the subtraction of blanks and the conversion of the signal to mean residue ellipticity (deg $\times$ cm$^2$ $\times$ dmol$^{-1}$ $\times$ res$^{-1}$).

## TBC1D31 and praja2 homology models

The wild-type FASTA sequence of human TBC1D31 was obtained from Uniprot website (Q96DN5) and submitted to I-TASSER server (Roy *et al*, 2010) in order to build the 3D homology model (Fig 2D). Similarly, the segment from W550 to D570 (praja2$_{W550-D570}$) of E3 ubiquitin-protein ligase praja2 (PJA2) human sequence was searched on Uniprot (O43164) and submitted to I-TASSER server. I-TASSER gave five homology models of both TBC1 and praja2$_{W550-D570}$. The quality of the predicted models was ranked based on their C-score, which is calculated based on the significance of threading template alignments and the convergence parameters of the structure assembly simulation. A higher value of the C-score signifies a model with high confidence. In our case, among the predicted models we selected the first model for both systems, having a C-score of $-0.99$ and $-0.57$ for TBC1D31 and praja2$_{W550-D570}$, respectively. Specifically, as concerns the TBC1D31 model, according to experimental data, the C-terminal alpha-helix region from Q941 to A970 was used for docking and molecular dynamics (MD) calculations.

## TBC1/praja2 protein–protein docking calculations

TBC1D31/praja2 protein–protein docking was performed in two steps. The first step was aimed to obtain a starting TBC1D31/praja2 complex conformation, and it was performed using AutoDock Vina ver. 1.0.2 software (Trott & Olson, 2010). Specifically, the grid box (size: 102 x 100 x 100) was built in order to include the whole alpha helix of TBC1D31 C-terminal segment Q941-A970, an exhaustiveness of 32 was used and 20 docking poses of W550-D570 praja2 were generated. In the second step, an AutoDock Vina docking pose refinement was carried out with FlexPepDock, a high-resolution peptide-protein docking protocol for the modelling of peptide-protein complexes implemented in the Rosetta framework (London *et al*, 2011). In particular, the best pose found by AutoDock Vina was submitted to FlexPepDock webserver, which confirmed the goodness of the docking sampling.

## TBC1D31/praja2 molecular dynamics simulation

The TBC1D31/praja2 FlexPepDock refined pose was submitted to 2μs-long MD simulation using the AMBER16 suite. The complex was firstly parameterized with the *LEaP* module of AmberTools16 suite, using the *ff14SB* force field (Maier *et al*, 2015). The system was immersed in a pre-equilibrated octahedral box of TIP3P water molecules and neutralized by adding Na$^+$ and Cl$^-$ counterions. The final system, of about 22,000 atoms, was minimized in three steps using an energy gradient convergence criterion set to 0.01 kcal/mol Å$^2$ involving: (i) only the hydrogen atoms of the system (2,000 steps of steepest descent); (ii) hydrogen atoms, water molecules and counterions (4,000 steps of steepest descent); (iii) minimization of the whole system (10,000 steps of steepest descent). Subsequently, water, ions and protein side chains were thermalized in two equilibration steps: (i) 200 ps heating water and ions from 0 to 298 K with constant volume, restraining protein atoms; (ii) 800 ps of thermalization step with pressure control at

1 atm (NPT ensemble) of the whole system, without any restraint; (iii) additional 400 ps were performed in order to further equilibrate the system density in NPT ensemble. Finally, the 2 μs-long of production run was performed in NPT using a time step of 2 fs. MD's trajectory was analysed with VMD 1.9.4 (Humphrey *et al*, 1996). A cluster analysis of MD trajectory was conducted considering a praja2 peptide RMSD cut-off of 5.0 Å. All the images were rendered with UCSF Chimera (Pettersen *et al*, 2004).

## Protein network analysis

The protein–protein interaction sub-network has been obtained by using GeneMANIA (Mostafavi *et al*, 2008; Montojo *et al*, 2010). In particular, the network has been built using ODF1, PJA2 and TBC1D31 as input seeds and the GeneMANIA "Physical Interactions" catalogue as protein–protein interaction database. This catalogue is composed by all interactions reported in different experiments and collected in different databases such as BioGRID and PathwayCommons. Starting from the input proteins and the chosen set of protein–protein interaction networks, the GeneMANIA algorithm extracts a single association network, centred on the input proteins and summarizing the information from all the different networks. After the physical interaction network has been obtained, the subnetwork induced by all the first-order interactors of ODF1, PJA2 and TBC1D31 was computed by filtering out from the network all the nodes not having any direct (physical) interaction with the input proteins. Finally, the set of genes associated with the proteins in the final network was functionally annotated on Gene Ontology terms using the g:GOst module of the gProfiler (cit) toolset, with G:SCS as multiple test correction method and 0.05 as *P*-value threshold.

## Medaka stocks

The Cab-strain of wild-type Medaka fish (*Oryzias latipes*) was maintained following standard conditions (i.e. 12 h/12 h dark/light conditions at 27°C). Embryos were staged according to (Avellino *et al*, 2013). All studies on fish were conducted in strict accordance with the Institutional Guidelines for animal research and approved by the Italian Ministry of Health; Department of Public Health, Animal Health, Nutrition and Food Safety in accordance to the law on animal experimentation (D. Lgs. 26/2014). Furthermore, all animal treatments were reviewed and approved in advance by the Ethics Committee at the TIGEM Institute, (Pozzuoli, NA), Italy.

## Sequence analysis

The available medaka *Ol-TBC1D31* (ENSORLT00000007876.1) at chr16:11638220-11645068 genomic sequence was retrieved from public databases (http://genome-euro.ucsc.edu/) and aligned with human *TBC1D31* transcript (ENST00000287380.6) to identify exons based on sequence.

## mRNA and MO injection of Medaka embryos

In vitro synthesis of human *TBC1D31*, wild-type *OFD1*, *OFD1$_{S735A}$*, *OFD1$_{S735D}$* and *hpraja2rm* mRNAs were performed following manufacture's instruction (Porpora *et al*, 2018). mRNAs were injected at 25–100 ng/μl to observe dose-dependent phenotypes; selected

working concentrations were 50 ng/µl for single injections and 25 ng/µl for combined injections. A morpholino (Mo; Gene Tools LLC, Oregon, USA) was designed against the splicing donor of exon 3 (Mo-*Ol-TBC1D31*: 5′-CACCAGACGAAATCTACTCCAAGTT-3') of the medaka orthologous of the *TBC1D31* gene. Mo-*Ol-TBC1D31* was injected at 0.15 mM concentration into one blastomere at the one/two-cell stage. Off-target effects of the morpholino injections were excluded by repeated experiments with control morpholino or by co-injection with a p53 morpholino (Mo-p53: 5′-CGGGAATCG-CACCGACAACAATACG-3′) as described (Porpora *et al*, 2018).

### Whole-mount immunostaining

Whole-mount immunostaining was performed and photographed, as described (Conte *et al*, 2010). Embryos at Stage 24 were fixed in 4% paraformaldehyde, 2X phosphate-buffered saline (PBS) and 0.1% Tween-20. The fixed embryos were detached from chorion and washed with PTW 1X. Embryos were digested 7 min with 10 g/ml proteinase K and washed twofold with 2 mg/ml glycine/PTW 1X. The samples were fixed 20 min in 4% paraformaldehyde, 2X phosphate-buffered saline (PBS) and 0.1% Tween-20, washed with PTW 1X and then incubated for 2 h in FBS 1%/PTW 1X, at room temperature. The embryos were incubated with mouse anti-acetylated α-tubulin antibody 1:400 (6- 11B-1; Sigma-Aldrich, St Louis, MO, USA), overnight at 4°C. The samples were washed with PTW 1X, incubated with the secondary antibody, Alexa-488 goat anti-mouse IgG (Thermo Fisher), then with DAPI. Finally, the embryos were placed in glycerol 100%.

### Real-time RT–PCR

Transcriptional levels of Sonic Hedgehog signalling pathway associated-genes in human samples were analysed by real-time RT–PCR. Total mRNA was extracted with TRIzol reagent (Sigma) according to the manufacturer protocol. 1µg of total mRNA from each sample was retrotranscribed using Luna Script RT SuperMix Kit (New England Biolabs). Real-Time PCR was performed using SYBR Green Master Mix (Biorad) using the following primers: Gli2 primer 1: 5'-CAA CGCCTACTCTCCCAGAC-3'; Gli2 primer 2: 5'-GAGCCTTGATGTA CTGTACCAC-3'; TBC1D31 primer 1: CAAGATATGGCACCGCAAGC; TBC1D31 primer 2: TAAAGGCCAGAGCTGTGCAA. Each reaction was carried in triplicate using 25 ng of cDNA in 20µl. As internal control, 18S ribosomal RNA expression levels were used. Melting curves were obtained by increasing the temperature from 60 to 95°C with a temperature transition rate of $0.5°C\,s^{-1}$. Melting curves of final PCR products were analysed (OpticonMonitor 3 Bio-Rad).

## Data availability

This study includes no data deposited in external repositories.

**Expanded View** for this article is available online.

## Acknowledgements
This work was supported by the "Associazione Italiana per la Ricerca sul Cancro" (AIRC, IG 2018-ID22062), the Italian Ministry of University and Research (PON 2018, Per. Med. Net) and the European Regional Development Fund-POR Campania FESR 2014/2020 (Rare. Plat. Net) to AF. FC was supported by fellowships from the Italian Ministry of University and Research (PON 2018, Per. Med. Net) and PRIN2017 (2017237P5x). BC and AF acknowledges support from the European Regional Development Fund-POR Campania FESR 2014/2020 (Satin). M.K. acknowledges the promotion programme for young scientist at the University of Innsbruck (project no: 316826). ES was supported by Austrian Science Fund (P30441, P32960). Thanks to Dr Laurence Pelletier for providing GFP-TBC1D31 vector. We thank Francesco Salierno for technical assistance at TIGEM Medaka Fish Facility. Special Thanks to Max Gottesman Columbia University for critical reading of the manuscript and Luca Lignitto for the initial characterization of the TBC1D31 clone from the yeast two-hybrid screening.

## Author contributions
Experiments and data analysis: ES, FC, RDD, LR, DI, LP, BF, FM, GG, GS, E Stefan, MM, AR and CG; Mass spectrometric experiments and data analysis: OT-Q, MK and AP; Data analysis: BF; Experiments conception, data supervision and analysis: AF, EP, BC and IC; Manuscript writing: AF.

## Conflict of interest
The authors declare that they have no conflict of interest.

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
