## [Review Process File · The EMBO Journal]

The TBC1D31/praja2 complex controls primary ciliogenesis through PKA-directed OFD1 ubiquitylation

Emanuela Senatore, Francesco Chiuso, Laura Rinaldi, Daniela Intartaglia, Rossella Delle Donne, Emilia Pedone, Bruno Catalanotti, Luciano Pirone, Bianca Fiorillo, Federica Moraca, Giuliana Giamundo, Giovanni Scala, Andrea Raffeiner, Omar Torres-Quesada, Eduard Stefan, Marcel Kwiatkowski, Alienke van Pijkeren, Manuela Morleo, Brunella Franco, Corrado Garbi, Ivan Conte, and Antonio Feliciello

DOI: 10.15252/embj.2020106503

Corresponding author(s): Antonio Feliciello (feliciel@unina.it)

Review Timeline:

Submission Date:	14th Aug 20
Editorial Decision:	17th Sep 20
Revision Received:	21st Dec 20
Editorial Decision:	1st Feb 21
Revision Received:	3rd Feb 21
Accepted:	11th Feb 21

Editor: Elisabetta Argenzio

Transaction Report:

Thank you for submitting your manuscript entitled "The TBC1D31/praja2 complex controls primary ciliogenesis through PKA-directed OFD1 ubiquitination" (EMBOJ-2020-106503) to The EMBO Journal. Your study has been sent to three referees for evaluation and we have now received reports from them, which are enclosed below for your information.

As you can see, while the referees find your work potentially interesting, they also raise several major points that need to be addressed before they can support publication in The EMBO Journal. In particular, referee #1 and #2 are concerned that most of the results are based on protein overexpression. In addition, referee #1 stresses that data have to be reproduced in more appropriate cell lines and that the relevance of these findings needs to be further investigated in vivo.

We agree with the referees that these are important points and addressing these as well as all other reviewers' criticisms will be essential to pursue publication of this study in The EMBO Journal. Strong support from the referees would also be needed for publication here. Given the overall interest of your study, I would like to invite you to submit a revised version of the manuscript according to the referees' requests. I should add that it is The EMBO Journal policy to allow only a single round of revision, and acceptance of your manuscript will therefore depend on the completeness of your responses in this revised version.

I realize that addressing all the referees' criticisms will require time and additional efforts that might also be technically challenging. I would therefore understand if you were to choose not to undergo an extensive revision here and rather pursue a submission elsewhere, in which case please inform us about your decision at your earliest convenience.

Referee #1:

In this manuscript Senatore et al. investigate the role of TBC1D31 and praja2 as regulators of OFD1 through PKA. Mutations in OFD1 lead to oro-facial digital type 1 syndrome (OFD1) which is an x-linked ciliopathy. OFD1 is a component of the centrosome / basal body and has been linked to ciliary dynamics and signaling. The at hand story starts with the identification of TBC1D31 as a novel interacting protein of the E3 ubiquitin ligase praja2 by a Y2H screen. The interaction is mapped and further characterized. TBC1D31 and praja2 are found to be co-localized at the centrosome and praja's localization depends on the expression of TBC1D31. Next, both TBC1D31 and praja2 appear to co-precipitate and co-localize with the ciliopathy protein OFD1. Further analyses result in a model, in which a GPCR/cAMP activation of PKA leads to phosphorylation and subsequent ubiquitination/degradation of OFD1 via praja. The authors suggest, that this pathway is essential for ciliogenesis and that phosphorylation deficient OFD1 affects cilia morphology. Ultimately, they investigate the role of TBC1D31 and mutant OFD1 in Medaka.

In my opinion, this story is interesting and alludes to a potentially important molecular mechanism underlying OFD1 function and regulation in cilia biology. However, most of the data presented is based on overexpression in HEK293 cells. In addition, the data on ciliogenesis, localization and disassembly is primarily based on HEK293 cells, too. They are of course easy to maintain, to handle and to transfect. But these undifferentiated immortalized cells, most likely of unclear origin and with unknown passage number, are not a good model to study cilia biology. In Fig. 6 controls have between 10% and 30% ciliated cells, which already indicates the instability of the system. We typically do not observe any normal cilia in 'our' HEK cells. Therefore, also for the sake of reproducibility I would highly recommend to include 1 or 2 additional, well characterized cell lines with a real high percentage of ciliated cells into this study. Moreover, more in vivo data should be included.

Major comments:

1. Fig.1 shows the interaction of TBC1D31 and praja2 using overexpression of GFP and FLAG tagged proteins. Since antibodies are available, co-immunoprecipitation experiments should be performed both from cell lysates as well as from relevant tissues. Does Fig. 1E use lysates from untransfected cells? Here, showing full blots would be helpful to demonstrate the quality of the antibodies used.
2. Suppl. Fig. 1: siRNA is used to control the TBC1D31 antibody. This siRNA should be controlled independently as well, e.g. by qPCR. This will be an important control for later figures as well, e.g. Fig. 6B.
3. More data should be added on the complex of praja2, TBC1D31 and OFD1. How exactly was the blot for Fig. 3C done? Are these the same samples run on separate gels or lanes? Or have the membranes been striped two times? All three proteins run exactly on the same size. Exchanging the tags between OFD and TBC (then GFP.OFD and Flag.TBC....) would perhaps solve this problem here. In addition, using endogenous proteins (and perhaps tissue lysates) rather than overexpression in cells would be more convincing.
4. The resolution of the IF pictures is quite low. TBC1D31, praja2 and OFD1 seem to co-localize at

the centrosome (Fig 3E), but OFD1 does not completely overlap with the others. Using superresolution imaging such as STED would be helpful to define the exact localizations. In addition, co-stainings with centrosome markers would be good.

5. Figure 4 identifies residue S735 of OFD1 as PKA target. Phosphorylation is shown with a phospho-substrate antibody using an in vitro-kinase reaction and overexpression and IP in HEK cells. However, there is no evidence that this phosphorylation occurs in vivo or is affected in OFD1 or related cilioapthies. Has this phosphorylation been observed before? Does it occur in vivo? Phosphorylation should be shown with endogenous proteins. In addition, it might be good to use a method independent of the substrate antibody to show phosphorylation.

6. The authors state in the abstract, that "this pathway is essential for ciliogenesis" referring to cAMP-PKA-OFD-praja1/TBC1D31. However, they only provide data on TBC1D31 KD affects ciliogenesis and that OFD1 mutant overexpression might affect cilia. This is not necessarily linked. While the paper suggests a role for the cAMP-PKA-OFD-praja1/TBC1D31 module or pathway, this is not demonstrated experimentally. The Medaka data does also not provide a direct link between TBC1D31 and OFD1. A more severe phenotype in the combination of TBC1D31 KD with OFD S735A mRNA does not prove any direct link, especially since OFD1 WT has not effect here.

Minor comments:

1. The GST/GST:TBC... levels appear to be quite unequal in Fig 1D.
2. The quality of Fig. 2A should be improved.
3. Fig. 4B: The exact size of GST.OFD1 should be indicated. What are the additional bands? An anti-GST blot could be provided. Fig. 4C lower panel / Fig 4E middle panel: Are these lysates or precipitates? Is expression in the lysates equal?
4. Fig. 5 D;G;I: error bars should be removed from the controls that were set to "1".
5. Sequences of the control siRNAs should be included.

Referee #2:

Senatore et al. presented evidence for a new function of the TBC1D31/praja2 complex in primary ciliogenesis through PKA-directed OFD1 ubiquitination. In their manuscript the authors showed that TBC1D31 localizes at centrosomal region and serves as a scaffold to recruit praja2 and PKA to regulate ciliogenesis, PKA and regulates ciliogenesis by phosphorylating OFD1, and the phosphorylation of OFD1 promotes the praja2-UPS mediated proteolysis of OFD1. OFD1 is a key regulator of cilia formation and ciliary signaling. Although the autophagy-mediated degradation of OFD1 is reported, the ubiquitin-proteasome mediated degradation of OFD1 is unknown. The results are interesting and potentially important. However, there are several concerns about this study, which should be addressed before publication.

Major concerns

-There are concerns about several overexpression experiments. Overexpression of OFD1 and several other centriolar satellite proteins likely cause protein aggregation rather than functional distribution. In this study, overexpression of OFD1 displayed only centrosome staining without much centriolar satellite staining, it will be more convincing to show endogenous staining of these proteins.

- In Figure 1, praja2 antibody showed TBC1D31 dependent centrosomal localization. The specificity of this antibody should be validated by KD or KO in the staining experiment.

- Can the authors provide evidence to show that PKA colocalizes at centrosome with praja2 and OFD1?
- It has been known that OFD1 is degraded by autophagy during serum starvation. Can the authors compare the role of autophagy and proteasome pathways in ciliogenesis?
- The authors showed that OFD1 can be phosphorylated. During serum starvation or other stresses, does the level of OFD1 phosphorylation alter upon stress? A specific phosphor-antibody of OFD1 will give more insight information.
- What's the site of OFD1 that is important for the poly-UB modification?
- In Figure 5, it is necessary to measure the half-life of OFD1 upon praja silencing or FSK treatment. Also the protein level difference could be due to degradation or protein synthesis, which is not well separated in this study.
- In Figure 6D, the acetylated tubulin is discontinuous in cells expressing OFD1 S375 mutant. What is the nature of this defect? The authors should examine the ciliary membrane integrity using ARL13b antibody or perform EM analysis to visualize the details of ciliary defects.

Minor concerns

- In most of the experiments, the cell line used was not specified.
- Germline inactivating mutations of OFD1 cause the Oral-Facial-Digital type I (OFDI) syndrome, syndrome, a developmental disorder usually characterized by typical oral-facial-digital malformations, renal cystic disease and central nervous system involvement (Bruel, Franco et al., 2017, Macca & Franco, 2009a).
There are two "syndrome", is this a typo?

Referee #3:

In the function of primary cilia, GPCR signaling and ubiquitination play important roles. However, it is not known how they are connected each other. Senatore and his colleagues identified a complex composed of TBC1D31, the E3 ligase praja2, protein kinase A and OFDI, a ciliopathy causative gene product, and characterized this complex by immunoprecipitation, immunofluorescent microscopy and genetic engineering. They demonstrated that the complex is located at the basal body and its defect causes abnormality of ciliogenesis in cultured cells as well as of development in medaka fish. The experimental data are organized well to examine their hypothesis that this complex connects GPCR signaling and ubiquitination and thus is key for ciliogenesis, and they proved it. This work is a milestone of the field of primary cilia and will evoke further studies such as high resolution structure of this complex and precise localization of it in the basal body. This reviewer supports publication of this manuscript after minor revision.

Minor points:

1. While the figures are beautifully organized, the figure captions have rooms of improvement. In the current manuscript, the captions from the most figures are written in the similar way as Methods and contain many detailed experimental descriptions. The caption of each panel can be rewritten to have short titles (as correctly done in Fig.7A-D, but not others), specific explanation of individual items (such as gel lines) and definition of indications (such as arrows in Fig.6G). Experimental details

should be moved to the Methods section.

2. Three diagram panels (Fig.3D, Fig.4G, Fig.7E) are not explained enough in the caption. (Fig.1H is self-explanatory)

3. Some statements in Introduction miss references, but will be benefited by appropriate references. For example: "OFD1 gene encodes centriolar and PCM proteins"; "orphan GPCR and AC within the cilium suggested locally negated cAMP microdomains directly controls PKA ...".

4. p.11 line7: wild-type -> wild-type

5 p.14 line2 "compartmentalized cAMP signalling": what kind of compartment do the authors mean?

6. Fig.4 "three independent experiments": did the authors cultured cells three times to run the same experiment, or run the gel three times from the same preparation?

REBUTTAL LETTER TO THE REVIEWERS
EMBOJ-2020-106503

Referee #1:

In this manuscript Senatore et al. investigate the role of TBC1D31 and praja2 as regulators of OFD1 through PKA. Mutations in OFD1 lead to oro-facial digital type 1 syndrome (OFD1) which is an x-linked ciliopathy. OFD1 is a component of the centrosome / basal body and has been linked to ciliary dynamics and signaling. The at hand story starts with the identification of TBC1D31 as a novel interacting protein of the E3 ubiquitin ligase praja2 by a Y2H screen. The interaction is mapped and further characterized. TBC1D31 and praja2 are found to be co-localized at the centrosome and praja's localization depends on the expression of TBC1D31. Next, both TBC1D31 and praja2 appear to co-precipitate and co-localize with the ciliopathy protein OFD1. Further analyses result in a model, in which a GPCR/cAMP activation of PKA leads to phosphorylation and subsequent ubiquitination/degradation of OFD1 via praja. The authors suggest, that this pathway is essential for ciliogenesis and that phosphorylation deficient OFD1 affects cilia morphology. Ultimately, they investigate the role of TBC1D31 and mutant OFD1 in Medaka.

In my opinion, this story is interesting and alludes to a potentially important molecular mechanism underlying OFD1 function and regulation in cilia biology. However, most of the data presented is based on overexpression in HEK293 cells. In addition, the data on ciliogenesis, localization and disassembly is primarily based on HEK293 cells, too. They are of course easy to maintain, to handle and to transfect. But these undifferentiated immortalized cells, most likely of unclear origin and with unknown passage number, are not a good model to study cilia biology. In Fig. 6 controls have between 10% and 30% ciliated cells, which already indicates the instability of the system. We typically do not observe any normal cilia in 'our' HEK cells. Therefore, also for the sake of reproducibility I would highly recommend to include 1 or 2 additional, well characterized cell lines with a real high percentage of ciliated cells into this study. Moreover, more in vivo data should be included.

R. *We wish to thank the Reviewer to find our results 'interesting and potentially important mechanism underlying OFD1 function and regulation in cilia biology'. We agree with the Reviewer comments that most of the experiments were made using overexpressed proteins and the cell line (HEK293) used may not be sufficient to support the role of PKA-UPS-OFD1 axis in cilium biology. Accordingly, we have replicated the experiments in other cell lines, performed experiments on endogenous proteins and strengthen the relevance of the findings in vivo. Please, see below for the changes made.*

Major comments:

1. Fig.1 shows the interaction of TBC1D31 and praja2 using overexpression of GFP and FLAG tagged proteins. Since antibodies are available, co-immunoprecipitation experiments should be performed both from cell lysates as well as from relevant tissues. Does Fig. 1E use lysates from untransfected cells? Here, showing full blots would be helpful to demonstrate the quality of the antibodies used.

R. *We have performed coimmunoprecipitation experiments using endogenous proteins from cell lysates, as well as from mouse brain. The results support the presence of an endogenous TBC131/praja2 complex in vivo (New Fig. 3D). For space limitations, we have included in the main figure the relevant parts of the blots. The full blots showing the specificity of the immunoreactive signals are shown in the 'Source data' section.*

2. Suppl. Fig. 1: siRNA is used to control the TBC1D31 antibody. This siRNA should be controlled independently as well, e.g. by qPCR. This will be an important control for later figures as well, e.g. Fig. 6B.

R. *We have now included the appropriate control for siRNA experiments by qPCR analysis on the TBC1D31 mRNA (new Fig. EVID).*

3. More data should be added on the complex of praja2, TBC1D31 and OFD1. How exactly was the blot for Fig. 3C done? Are these the same samples run on separate gels or lanes? Or have the membranes been striped two times? All three proteins run exactly on the same size. Exchanging the tags between OFD and TBC (then GFP.OFD and Flag.TBC....) would perhaps solve this problem here. In addition, using endogenous proteins (and perhaps tissue lysates) rather than overexpression in cells would be more convincing.

R. *The immunoblots for the three proteins are from the same samples that were run on the same gel. The blots were stripped and reprobed. The full blots showing the specificity of the immunoreactive signals are shown in the 'Source data' section.*

4. The resolution of the IF pictures is quite low. TBC1D31, praja2 and OFD1 seem to co-localize at the centrosome (Fig 3E), but OFD1 does not completely overlap with the others. Using superresolution imaging such as STED would be helpful to define the exact localizations. In addition, co-stainings with centrosome markers would be good.

R. *We thank the Reviewer for these criticisms and suggestion. To better support the localization data, we have repeated the triple immunostaining analyses monitoring the localization of endogenous TBC1D31, praja2 and OFD1 and including a centrosomal marker (γ -tubulin) (Please, see new Fig. 3E, Fig. EV2A-C).*

5. Figure 4 identifies residue S735 of OFD1 as PKA target. Phosphorylation is shown with a phospho-substrate antibody using an in vitro-kinase reaction and overexpression and IP in HEK cells. However, there is no evidence that this phosphorylation occurs in vivo or is affected in OFD1 or related cilioopathies. Has this phosphorylation been observed before? Does it occur in vivo? Phosphorylation should be shown with endogenous proteins. In addition, it might be good to use a method independent of the substrate antibody to show phosphorylation.

R. *We agree with the Reviewer. To address this important criticism, first we monitored phosphorylation of endogenous OFD1 in cells stimulated with FSK. As shown in the new Fig. EV3A, cAMP stimulation markedly induced OFD1 phosphorylation. To directly address whether endogenous OFD1 is, indeed, phosphorylated at S735 by PKA, we decided to perform MS analyses of affinity isolated OFD1 complexes from Hela cells expressing PKAc subunit. The endogenous phosphorylation site S735 was identified by mass spectrometry (new Fig. EV3B-D). Furthermore, this site has been identified previously. For reference, please see below:*

<https://www.phosphosite.org/siteGroupAction.action?id=480374&protOrg=12569&showAllSites=true&showHTPRefsOnly=true>

6. The authors state in the abstract, that "this pathway is essential for ciliogenesis" referring to cAMP-PKA-OFD-praja1/TBC1D31. However, they only provide data on TBC1D31 KD affects ciliogenesis and that OFD1 mutant overexpression might affect cilia. This is not necessarily linked. While the paper suggests a role for the cAMP-PKA-OFD-praja1/TBC1D31 module or pathway, this is not demonstrated experimentally. The Medaka data does also not provide a direct link between TBC1D31 and OFD1. A more severe phenotype in the combination of TBC1D31 KD with OFD S735A mRNA does not prove any direct link, especially since OFD1 WT has not effect here.

R. We thank the Reviewer for his/her important comments. Indeed, in the previous version of the manuscript we have characterized the role of *OFD1*_{S735A} mutant in ciliogenesis and medaka fish development. In the current revised version, we show additional experiments addressing the concerns raised and modified the manuscript accordingly. In particular, to better characterize the *TBC1D31*•*praja2*•*PKA*•*OFD1* axis in vivo, we used an *OFD1*_{S735D} mutant mimicking a constitutively phosphorylated form of *OFD1* and a dominant negative variant of *praja2* (*praja2rm*) that is unable to ubiquitylates *OFD1*. We reasoned that overexpression of *OFD1*_{S735D} in the presence of *praja2rm* mutant might lead to a rescue of ciliogenesis in *TBC1D31* morphants. Consistent with previous results and in agreement to our model, the new experiments show that only co-expressing *praja2rm* and *OFD1*_{S735D} mutant induces the rescue of both ciliogenesis and embryo development in *TBC1D31* morphants. We think that the results of these assays now better support the notion that *PKA-OFD1-praja2/TBC1D31* act in a complex to control ciliogenesis. These results are described in the 'Results' section and are shown in the **new Fig. 7A-C** and **Appendix Fig. S5A-C**.

Minor comments:

1. The GST/GST:TBC... levels appear to be quite unequal in Fig 1D.

R. We have repeated the GST-pull down experiments. Please, see the **new Fig. 1D**.

2. The quality of Fig. 2A should be improved.

R. The full size of the blot is shown in the 'Source data' section.

3. Fig. 4B: The exact size of GST.OFD1 should be indicated. What are the additional bands? An anti-GST blot could be provided.

R. We have indicated the exact size of GST-OFD1. Moreover, the additional bands visible with the Ponceau staining are degradative products of GST-OFD1 that often appear during the purification steps of the fusions from bacterial lysates. These products are now marked by an asterisk.

Fig. 4C lower panel / Fig 4E middle panel: Are these lysates or precipitates? Is expression in the lysates equal?

R. The experiments shown in Fig. 4C and 4E refer to flag-Ip samples. The expression of the transgenes was quite similar among the experimental groups.

4. Fig. 5 D;G;I: error bars should be removed from the controls that were set to "1".

R. Sorry for the mistake. It occurred during the preparation of the **Fig. 5D; G; I**. We have now removed the error bars from the controls.

5. Sequences of the control siRNAs should be included.

R. We have now included in the Methods the appropriate reference for control siRNAs.

Referee #2:

Senatore et al. presented evidence for a new function of the TBC1D31/praja2 complex in primary ciliogenesis through PKA-directed OFD1 ubiquitination. In their manuscript the authors showed that TBC1D31 localizes at centrosomal region and serves as a scaffold to recruit praja2 and PKA to regulate ciliogenesis, PKA and regulates ciliogenesis by phosphorylating OFD1, and the phosphorylation of OFD1 promotes the praja2-UPS mediated proteolysis of OFD1. OFD1 is a key regulator of cilia formation and ciliary signaling. Although the autophagy-mediated degradation of OFD1 is reported, the ubiquitin-proteasome mediated degradation of OFD1 is unknown. The results are interesting and potentially important. However, there are several concerns about this study, which should be addressed before publication.

R. *We thank the Reviewer for his/her comments on the manuscript and in finding the results 'interesting and potentially important'. We have made all the suggested changes and revised the manuscript and figures accordingly.*

Major concerns

- There are concerns about several overexpression experiments. Overexpression of OFD1 and several other centriolar satellite proteins likely cause protein aggregation rather than functional distribution. In this study, overexpression of OFD1 displayed only centrosome staining without much centriolar satellite staining, it will be more convincing to show endogenous staining of these proteins.

R. *We thank the Reviewer for these criticisms and suggestion. To better support the localization data, we have repeated the triple immunostaining analyses monitoring the localization of endogenous TBC1D31, praja2 and OFD1 and including γ -tubulin staining as a centrosome marker (Please, see new Fig. 3E, Fig. EV2A-C).*

- In Figure 1, praja2 antibody showed TBC1D31 dependent centrosomal localization. The specificity of this antibody should be validated by KD or KO in the staining experiment.

R. *We have now included the praja2 immunofluorescence in siRNAs-transfected cells (new Fig. EV1B).*

- Can the authors provide evidence to show that PKA colocalizes at centrosome with praja2 and OFD1?

R. *As suggested by the Reviewer, we analyzed the colocalization of PKA, praja2 and OFD1. The new Figure EV2C shows that PKAc subunit partly colocalizes with TBC1D31 and OFD1.*

- It has been known that OFD1 is degraded by autophagy during serum starvation. Can the authors compare the role of autophagy and proteasome pathways in ciliogenesis?

R. *The ubiquitin proteasome system and the autophagy pathway have a major and complex role in primary ciliogenesis. However, whether autophagy and UPS promote or inhibit ciliogenesis is still debated (Lam, Cloonan et al., J.C.I. 123, 2013; Pampliega, Orhon et al., Nature 502, 2013; Shearer & Saunders, 2016). Nevertheless, as suggested by the Reviewer, we monitored primary cilia in cells treated with an inhibitor of autophagy (bafilomycin) or with a proteasome inhibitor (MG132). While we did not observe significant changes in the number of cilia in bafilomycin-treated cells, we indeed found that inhibition of the proteasome activity markedly reduced the number of ciliated cells (Appendix Fig. S6A-B).*

- The authors showed that OFD1 can be phosphorylated. During serum starvation or other stresses,

does the level of OFD1 phosphorylation alter upon stress? A specific phosphor-antibody of OFD1 will give more insight information.

R. *Experiments aimed to analyze the phosphorylation of OFD1 by cAMP stimulation were performed in cells deprived of serum for 24 hours. As shown in the immunoblots from Fig. 4C and E, and Fig. EV3A, under basal conditions phosphorylation of OFD1 was very low and it can be markedly induced by FSK. However, we cannot exclude the possibility that other stress-induced kinases might phosphorylate OFD1 under different experimental conditions. As for phosphor-antibody of OFD1, at moment this is not commercially available nor used from other laboratories. Nevertheless, to directly address whether endogenous OFD1 is, indeed, phosphorylated at S735 by PKA, we decided to perform MS analyses of affinity isolated OFD1 complexes from Hela cells expressing PKAc subunit. Indeed, the endogenous phosphorylation site S735 was identified by mass spectrometry (new Fig. EV3B-D). Furthermore, this site has been identified previously. For reference, please see below:*

<https://www.phosphosite.org/siteGroupAction.action?id=480374&protOrg=12569&showAllSites=true&showHTPRefsOnly=true>

- What's the site of OFD1 that is important for the poly-UB modification?

R. *This is a really interesting question. In the published databases, OFD1 appeared to be a highly ubiquitylated protein and different lysine residues have been identified as acceptor sites for ubiquitin moieties. We think that this aspect needs substantial work and it could be object for future studies. For reference, please see below:*

<https://www.phosphosite.org/siteGroupAction.action?id=480374&protOrg=12569&showAllSites=true&showHTPRefsOnly=true>

- In Figure 5, it is necessary to measure the half-life of OFD1 upon praja silencing or FSK treatment. Also the protein level difference could be due to degradation or protein synthesis, which is not well separated in this study.

R. *We have now better addressed this aspect. Specifically, we show that the treatment of serum-deprived cells with cycloheximide, an inhibitor of protein synthesis, did not affect the bulk levels of OFD1 suggesting that the protein is quite stable. In contrast, FSK stimulation significantly reduced the half-life of OFD1 (Fig. EV4A and B), implying that cAMP is promoting proteolysis of OFD1. We wish to note that the experiments aimed to define the role of cAMP signaling in OFD1 proteolysis were all performed in the presence of cycloheximide.*

- In Figure 6D, the acetylated tubulin is discontinuous in cells expressing OFD1 S375 mutant. What is the nature of this defect? The authors should examine the ciliary membrane integrity using ARL13b antibody or perform EM analysis to visualize the details of ciliary defects.

R. *As suggested by the Reviewer, we have repeated the experiments with the OFD1S735A mutant monitoring cilium morphology with the ARL13B antibody (New Fig. 6C, lower panels).*

Minor concerns

- In most of the experiments, the cell line used was not specified.

R. *We have now added this information in the text.*

- Germline inactivating mutations of OFD1 cause the Oral-Facial-Digital type I (OFDI) syndrome, syndrome, a developmental disorder usually characterized by typical oral-facial-digital malformations, renal cystic disease and central nervous system involvement (Bruel, Franco et al., 2017, Macca & Franco, 2009a).

There are two "syndrome", is this a typo?

R. *Sorry for the mistake. We have corrected the text, accordingly.*

Referee #3:

In the function of primary cilia, GPCR signaling and ubiquitination play important roles. However, it is not known how they are connected each other. Senatore and his colleagues identified a complex composed of TBC1D31, the E3 ligase praja2, protein kinase A and OFDI, a ciliopathy causative gene product, and characterized this complex by immunoprecipitation, immunofluorescent microscopy and genetic engineering. They demonstrated that the complex is located at the basal body and its defect causes abnormality of ciliogenesis in cultured cells as well as of development in medaka fish.

The experimental data are organized well to examine their hypothesis that this complex connects GPCR signaling and ubiquitination and thus is key for ciliogenesis, and they proved it. This work is a milestone of the field of primary cilia and will evoke further studies such as high resolution structure of this complex and precise localization of it in the basal body. This reviewer supports publication of this manuscript after minor revision.

R. *Many thanks to the Reviewer in finding our work 'well organized and a milestone of the field of primary cilia'. As suggested, we have modified the text and improved the quality of the manuscript.*

Minor points:

1. While the figures are beautifully organized, the figure captions have rooms of improvement. In the current manuscript, the captions from the most figures are written in the similar way as Methods and contain many detailed experimental descriptions. The caption of each panel can be rewritten to have short titles (as correctly done in Fig.7A-D, but not others), specific explanation of individual items (such as gel lines) and definition of indications (such as arrows in Fig.6G). Experimental details should be moved to the Methods section.

R. *As suggested by the Reviewer, we have rewritten panel captions, changed the legend titles and moved the experimental details in the Methods section.*

2. Three diagram panels (Fig.3D, Fig.4G, Fig.7E) are not explained enough in the caption. (Fig.1H is self-explanatory).

R. *We have now better described the caption to diagram panels of **new Fig. 3E, 4G, 7D**.*

3. Some statements in Introduction miss references, but will be benefited by appropriate references. For example: "OFD1 gene encodes centriolar and PCM proteins"; "orphan GPCR and AC within the cilium suggested locally negated cAMP microdomains directly controls PKA ...".

R. *Apologies for the missing information. We have now included the appropriate references in the text.*

4. p.11 line7: wild-ype -> wild-type

R. *Done.*

5 p.14 line2 "compartmentalized cAMP signalling": what kind of compartment do the authors mean?

R. *We have now changed 'compartmentalized cAMP signalling' to 'ciliary cAMP signaling' (p14, line 5).*

6. Fig.4 "three independent experiments": did the authors cultured cells three times to run the same experiment, or run the gel three times from the same preparation?

R. *When we refer to different experiments, we mean that independent experiments preparing different lysates and performing distinct immunoblot analyses were performed. In the figure we only show a representative set of experiments that gave similar results.*

2nd Revision - Editorial Decision

1st Feb 2021

Thank you for submitting your revised manuscript. The manuscript has now been sent back to referee #1 and #2, whose comments are appended below. As you will see, both reviewers find that their criticisms have been sufficiently addressed and recommend the study for publication.

However, there are few editorial issues concerning the text and the figures that I need you to address before we can officially accept your manuscript.

Referee #1:

In this revised manuscript my major concerns have been addressed by the authors. While I still feel that there are some limitations due to overexpression and the chosen cell lines, I support publication now.

Referee #2:

The authors have properly addressed my concerns.

3rd Revision - Editorial Decision

11th Feb 2021

I am pleased to inform you that your manuscript has been accepted for publication in The EMBO Journal.

Corresponding Author Name: Antonio Feliciello

Manuscript Number: EMBOJ-2020-106503